# Replacement of Dietary Carbohydrate with Protein versus Fat Differentially Alters Postprandial Circulating Hormones and Macronutrient Metabolism in Dogs

**DOI:** 10.3390/metabo14070373

**Published:** 2024-06-30

**Authors:** Matthew Irick Jackson

**Affiliations:** Hill’s Pet Nutrition, Topeka, KS 66617, USA; matthew_jackson@hillspet.com; Tel.: +1-785-286-8639

**Keywords:** canine, ketogenic, postprandial, macronutrients, metabolome

## Abstract

The effect of dietary macronutrients on fasting and postprandial responses was examined. Thirty-six healthy dogs were fed a high-carbohydrate (HiCHO) food once daily for 5 weeks, followed by randomization to either a high-protein, low-carbohydrate (PROT_LoCHO) or high-fat, low-carbohydrate (FAT_LoCHO) food for 5 weeks, then crossed over to the other LoCHO food for 5 weeks. Plasma samples were obtained at the end of each feeding period at timepoints before (0 h) and 2 h post-feeding. Apparent total circulating energy availability was assessed as a summation of the energetic contributions of measured glucose, β-hydroxybutyrate, triglycerides (TGs), non-esterified fatty acids (NEFAs), and fatty acids not from TGs or NEFAs. In both the fed and fasted states, there were increases in circulating apparent total energy availability after feeding the FAT_LoCHO food compared with the HiCHO or PROT_LoCHO foods. Changes from the postabsorptive to postprandial points in catabolic, anabolic, and signaling lipids all exhibited food effects. Consumption of either LoCHO food led to lower leptin/ghrelin ratios in the fasted state relative to the HiCHO food. The FAT_LoCHO food led to the highest postprandial levels of the incretins gastric inhibitory peptide and glucagon-like peptide-1, yet the lowest increases in insulin relative to the other foods. These findings provide information on how macronutrients can influence dietary energy processing and metabolic health.

## 1. Introduction

The metabolic response to a meal is a determinant of health and fitness in dogs. Canine obesity leads to increased postprandial levels of glucose, triglycerides (TG), and insulin relative to lean dogs [1]. Obesity in dogs is also associated with an increase in leptin following a meal [2] and with decreased postprandial response of dietary carnitine, perhaps due to altered fat metabolism [3]. Metabolic status may interact with the macronutrient composition of a meal to influence organ health and the efficacy of pharmaceutical intervention. Relative to lean dogs, obese dogs exhibit increased postprandial TGs and fasting glucocorticoid levels after consuming a high-fat carbohydrate-containing meal [4]. Postprandial hypertriglyceridemia observed in obese dogs is associated with increased pancreatic lipase immunoreactivity, a hallmark of pancreatic dysfunction [5]. Canine Cushing syndrome with diabetes also leads to postprandial hyperglycemia and glucose excursions [6]. Dietary starch type as well as meal frequency (e.g., number of postprandial-postabsorptive transitions per day) can influence glycemic control and thus insulin dosing regimens in diabetic dogs, with high-glycemic index starch increasing the requirement for insulin in the morning [7].

The size and fitness level of dogs likely plays a role in postprandial responses, since the dispositions of carbohydrate and protein (as amino acids) are not equal across metabolically active organs versus muscle tissue. A study of the tissue-specific dispositions of glucose and gluconeogenic amino acids after a mixed-macronutrient meal in dogs revealed that the canine liver assimilates approximately three times more gluconeogenic amino acids than glucose [8]. In contrast, non-splanchnic tissues such as muscle primarily take in glucose and not as many gluconeogenic amino acids as the liver [8]. Food form and specific nutrients may also impact the postprandial responses of macronutrient metabolites (e.g., glucose, fatty acids, ketone bodies, amino acids, and urea). The ranges for postprandial increases in blood urea nitrogen (BUN) in dogs differ by the form of the food, such that canned foods produce higher increases in BUN following a meal than do dry foods [9] and meat protein is a large driver of postprandial BUN increase [10]. With regard to specific nutrients, the long-chain omega-3 polyunsaturated fatty acid eicosapentaenoic acid (20:5n3; EPA) is reported to decrease postprandial levels of the ketone body fat metabolite beta-hydroxybutyrate (BHB) in an age-dependent manner in humans [11]. However, long-chain unsaturated fats may increase the ketogenicity of foods via activation of the transcription factor peroxisome proliferator-activated receptor (PPAR) α after conversion to endocannabinoid ethanolamides [12,13]. Further, in humans, consumption of saturated medium-chain triglyceride oils (MCT) can decrease postprandial circulating TGs relative to consumption of longer-chain saturated fats [14], and the combination of fish oil and MCT can decrease postprandial lipemia [15].

Many relevant biochemical markers are feasibly assessed in the fed state [16], and some reports examine the postprandial responses of dogs to single macronutrients or mixed meals. The postprandial response of dogs to foods rich in starch has been reported to be similar to that of humans, to the degree that a glycemic index may be approximated for canines [17]. Another study showed that increasing the carbohydrate energy in dog food to levels that are relevant to commercial maintenance foods increases postprandial glycemia [18]. Consumption of glucose itself shows that as a species dogs are metabolically adept at clearing this simple sugar in the postprandial state when compared to domesticated cats [19]. Twenty-four-hour glucose patterns have been shown to have a circadian dependence in dogs with diabetes [20]. The postprandial responses of circulating BUN in dogs consuming cooked, raw, and processed protein meals show that BUN can take 16–24 h to return to baseline [21]. The postprandial urinary creatinine:urinary urea nitrogen ratio in dogs is greater with low-protein foods [22]. The postprandial kinetics of TGs have been examined in dogs, and a half-life of TG-associated fatty acids was reported as 16 min [23]. The fasting levels of TGs do not appear to be related to peak postprandial levels in healthy dogs [24]; however, another study showed that fasting triglyceride levels in obese and overweight dogs were significantly associated with peak postprandial levels [5]. The postprandial responses of metabolic hormones have been assessed in dogs following bolus consumption of a single macronutrient; maltodextrin (a starch-like digestible carbohydrate) consumption increased circulating insulin while fat consumption (as lard) increased glucagon-like peptide 1 (GLP-1) [25]. In a separate study, provision of a high-protein food led to reduced postprandial leptin and mitigated the postprandial decrease in adiponectin [26].

A previous report comparing the fasting metabolic responses of dogs to a high-carbohydrate food relative to two low-carbohydrate foods was recently published [27]. In that study, one food contained a high-starch carbohydrate/moderate protein/moderate fat food, and the other two foods had starch carbohydrate replaced with either protein or fat. The primary finding was that the macronutrient composition of foods impacts the type and amount of available fasting (postabsorptive) circulating energy in dogs. However, there is still a gap in the existing literature related to the fasting-fed transition (postprandial) energetic and metabolic responses of dogs to dietary macronutrients. Furthermore, new information on the fasting and fed responses of circulating regulatory and counter-regulatory metabolic hormones could enhance the understanding of canine health and disease.

The aim of this investigation was to address this gap in the literature by documenting the fasting-to-fed changes in types of circulating energy (e.g., glucose, fatty acids, and ketone bodies), types of amino acids and fat metabolites, and metabolic hormones (incretins, (an)orexigenic, and regulatory/counter-regulatory) that occur with feeding high-carbohydrate or low-carbohydrate foods. More specifically, the aim was to parse the differential responses to two distinct low-carbohydrate foods, one with carbohydrate energy replaced by protein and the other with carbohydrate energy replaced with fat. The canine subjects were metabolically habituated to the foods and assessed in the fifth week of feeding, while the study controlled for health status, age, activity level, sex, and nutrient type. The results provide information on canine metabolism in a manner relevant to the management of canine diseases including endocrine disorders and cancer.

## 2. Materials and Methods

### 2.1. Study Foods and Analyses

The three foods in this study were high-carbohydrate (HiCHO, a typical standard adult maintenance food), high-protein and low-carbohydrate (PROT_LoCHO), and high-fat but low-carbohydrate (FAT_LoCHO; Table 1). As previously in [27], both low-carbohydrate foods contained less than 10% energy as carbohydrate. Foods differed from the prior study [27] in that the HiCHO and PROT_LoCHO foods were from different commercial lots and the FAT_LoCHO food contained the same nutrition parameters as before but was made from a different set of ingredients due to some ingredients being unavailable.

Similar to the preceding trial [27], only the FAT_LoCHO food contained added medium-chain fatty acids (MCF; e.g., C8:0 and C10:0) and long-chain n3 polyunsaturated fatty acids (PUFA; e.g., EPA and docosahexaenoic acid [DHA]) from ingredients rich in these nutrients (MCT and fish oil, respectively). The inclusion of the fish oil was to mitigate the potential of high-fat, low-carbohydrate foods to decrease circulating levels of n3 PUFA, as was observed in a rodent model [28]. The inclusion of MCT was to document postprandial increases in BHB resulting from food forms of MCT in dogs.

Nutrients in the foods, proximate analyses, and fatty acids were assessed as previously described [27]. Briefly, a commercial laboratory (Eurofins Scientific, Inc., Des Moines, IA, USA) determined the nutrients in the foods. Methods from the Association of Official Agricultural Chemists (AOAC) were used for proximate analyses: moisture—AOAC 930.15; protein—AOAC 2001.11; fat—AOAC 954.02; starch—AOAC 996.11; sugar profile and total sugars—AOAC 977.20; fiber—AOAC 962.09 [29]. Gas chromatography of methyl esters was used to measure fatty acids [30]. Sugars were summed from the individual analytic values for fructose, glucose, lactose, maltose, and sucrose, and total dietary fiber was summed from the analytic values for insoluble and soluble fiber.

### 2.2. Animals and Experimental Design

Thirty-six dogs (18 neutered males and 18 spayed females) participated in the study. Sixteen of the canine subjects (9 neutered males and 7 spayed females) also participated in the preceding trial using similar foods [27], which concluded 4 months prior to the start of the trial reported here. The sample size employed in the current study was the same as for this preceding study, as that sample size was sufficient to detect differences in fasting circulating energy. Subject participation in the two studies and animal signalment for the current trial are detailed in Appendix A. The mean ± standard error age at time of randomization was 7.14 ± 0.47 years, and body weight (BW) was 10.39 ± 0.27 kg. Dogs diagnosed with a chronic disease condition were excluded from the study.

The trial was a prospective, randomized crossover design to control for the effect of food order in the LoCHO foods. Dogs consumed the HiCHO food for 5 weeks, with collections in the last week of feeding. Dogs were then randomized into two groups (each n = 18) who were to receive the LoCHO foods in the opposite order. Thus, dogs were fed either the PROT_LoCHO or FAT_LoCHO foods for 5 weeks, with collections in the fifth week. Finally, the dogs were crossed over to the other LoCHO food they had not yet eaten and consumed that for 5 weeks with collections in the last week. Dogs were randomized based on sex, age, body weight, and breed such that there were no statistically significant differences in these parameters between groups. The metabolizable energy (ME) was continually adjusted to maintain the initial BW of the dogs at the study start, where the dogs’ activity factor was determined from historical caloric intakes and resting energy requirement = 70*(BW kg)^0.75^.

The study design allowed for normal socialization and enrichment activities for the dogs, which included daily group exercise in outdoor grassy runs that allowed for exposure to seasonal factors. While all pets had the opportunity to have exercise and interaction together in large groups (~20 dogs), they were pair-housed for sleeping arrangements. The dogs remained in their preferred housing arrangement during the trial, as previously determined by the colony veterinarian’s assessment of temperament and social interactions. Location as a confounder was controlled by including dogs from three separate buildings in the colony. Of the 36 dogs participating in the trial, there were only 6 dogs co-housed with another dog who was also participating in the trial (three pairs). Of these, two pairs included a dog from each of the food order groups and one pair was composed of dogs from group 2 (which received the FAT_LoCHO food first). Dogs were fed individually, once daily, in the morning (typically between 0800 and 0900 h) in individual feeding stations. The food intake amount (g/day) for each dog was collected through electronic feeders using a weight scale where each individually identified pet (through a radio frequency identification chip reader) was given access for 30 min to a controlled amount of food. Water was available ad libitum. Dogs were weighed weekly throughout the study. The study was carried out with approval from the Hill’s Pet Nutrition Institutional Animal Care and Use Committee (IACUC) and in accordance with Hill’s Global Animal Welfare Policy. Dogs were not subjected to any invasive procedures or those expected to cause pain, discomfort, or distress. Every dog received a complete physical exam and blood work to rule out systemic disease prior to and upon completion of the study. All dogs returned to the colony healthy after the study.

### 2.3. Sample Collection and Analyses

Fasted and 2-h postprandial blood collections were performed under an approved IACUC protocol (Protocol #: FP885.1.2.0-A-C-D-ADH-MULTI-112-MULTI). On collection days, dogs had fasted blood samples drawn approximately 23 h after their meal on the preceding morning and 30 min prior to their meal on the morning of the collection. These healthy canine subjects rarely take longer than 10 min to consume their day’s ration. The second collection occurred two hours after the food offering. Analyses of biochemical blood chemistry were performed on a COBAS c501 module (Roche Diagnostics Corporation, Indianapolis, IN, USA). Circulating BHB was analyzed by enzymatic reaction (IDEXX BioAnalytics Inc., Columbia, MO, USA). Gas chromatography of methyl esters with flame ionization detection was used to measure serum fatty acids [30]. Fatty acid elongation and desaturase enzymes were approximated from the ratios of relevant lipids [31]. Plasma metabolomics were performed by Metabolon (Morrisville, NC, USA), as previously described [32].

As previously [27], a composite of total apparent circulating metabolic energy availability was approximated from glucose, BHB, TGs, non-esterified fatty acids (NEFAs), and total fatty acids minus TGs (adjusted fatty acids) in a modified method [33]. Summing these four metrics gave a value for the total apparent circulating energy (kcal/L). Total fatty acids contained both NEFAs and fatty acids derived from complex lipids such as TGs, phospholipids, and cholesteryl esters. For adjusted fatty acids, the caloric contribution of TGs (fatty acids only) and NEFAs were subtracted to avoid double counting fat energy from TGs, since fatty acids from TGs would be accounted for by the assay for total fatty acids. Thus, apparent total circulating energy (kcal/L) = glucose (kcal/L) + BHB (kcal/L) + TGs (kcal/L) + adjusted fatty acids energy (kcal/L), where adjusted fatty acids (kcal/L) = total fatty acids (kcal/L) − TGs (kcal/L) − NEFAs (kcal/L). The values for the energy density of the individual components of total circulating energy were glucose (4 kcal/g), BHB (4.2 kcal/g), TGs (9 kcal/g), NEFAs (9 kcal/g), and adjusted fatty acids (9 kcal/g). These individual components (kcal/g) were multiplied by their respective measured levels in the blood (g/L) to arrive at a summation of total circulating energy (kcal/L). The energetic contributions of individual components (kcal/L) were then divided by the total circulating energy (kcal/L) to arrive at the percentage contribution of a given component to the total.

### 2.4. Statistical Analysis

Statistical analyses were conducted in JMP (Version 16.0. SAS Institute Inc., Cary, NC, USA, 1989–2022), including multivariate analysis of variance (MANOVA), the linear mixed model, and nonparametric dependent samples (paired) Wilcoxon signed-rank test. To determine if a multivariate class of metabolites differed by food type, MANOVA was performed using the Identity function to fit a model for each metabolite individually and then jointly test the models together. The largest of the MANOVA *p* values for Wilks’ Lambda, Pillai’s Trace, Hotelling-Lawley, and Roy’s Max Root is reported in the Appendix A. When *p* values for all of these MANOVA metrics were less than 0.05, the metabolite class was considered to be impacted by food type. The individual metabolites of multivariate classes were also summed into a single value to assess the broad impact of the food types on the classes. The overall effect of food type on biochemical analytes, levels of circulating energy, individual fatty acids, summed levels of metabolite classes, and hormones was determined by univariate linear mixed modeling. To account for the repeated measures design in this analysis, food was the main effect and subject was the random effect. To determine whether individual analytes differed between food types separately for the 0-h, 2-h, and delta (2 h–0 h) values, a Wilcoxon signed-rank test was performed for each pairwise combination of foods. Similarly, to determine whether a change occurred for a given analyte within a single food type when moving from the fasting-to-fed state, a Wilcoxon signed rank test was performed for each food using paired values at 0-h and 2-h timepoints for each subject. Bivariate relationships between analytes within a food treatment were assessed by Pearson correlation coefficient.

## 3. Results

### 3.1. Comparative Macronutrient and Lipid Composition of Foods: HiCHO, PROT_LoCHO, and FAT_LoCHO

The analyzed nutrient content of the foods is shown in Table 1. These foods largely paralleled the same compositions as those utilized in a separate trial published previously [27]. Predictably, the HiCHO food was by design the most replete in digestible carbohydrate (comprising starch and sugars), the PROT_LoCHO food was highest in protein, and the FAT_LoCHO food contained the greatest amount of fat. As in the preceding trial [27], the digestible carbohydrate mostly stemmed from starch rather than sugar, such that the ratio of starch to sugar was between 5- and 30-fold. The HiCHO and PROT_LoCHO foods were similar in total dietary fiber content, while the HiCHO and FAT_LoCHO foods were most alike in soluble fiber content.

### 3.2. Consumption of Energy and Nutrients Vis-à-Vis BW Maintenance

Intake of energy and nutrition are shown in Table 2. The ME ration was provided and continually adjusted to maintain the initial BW of the dogs at the study start. The mean ± SE BWs were as follows: HiCHO—10.39 ± 0.28 kg; LoCHO_PROT—10.37 ± 0.28 kg; LoCHO_FAT—10.35 ± 0.28 kg. There was no overall effect of food on BW (*p* = 0.314). In the current study, the energy intake (ME) required to maintain the dogs’ BW exhibited a significant food effect (*p* < 0.0001, Table 2). The calories required for BW maintenance were significantly lower when dogs were fed the PROT_LoCHO than either HiCHO or FAT_LoCHO foods (*p* < 0.0001). The amount of calories required for BW maintenance when dogs consumed the FAT_LoCHO trended towards being higher relative to when eating the HiCHO food but did not reach significance using a two-tailed *t*-test (*p* = 0.074). Given that there were previous data to support the idea that more calories would be required to maintain BW when dogs ate the FAT_LoCHO food relative to the HiCHO food, a one-tailed directional test might be more appropriate and provided a significant result of *p* = 0.037). There were significant differences across all comparisons among each of the three foods for the intake of protein, fat, saturated fatty acids, monounsaturated fatty acids, n3 PUFA (but not n6 PUFA), several fatty acids, carbohydrate (including total, starch, and sugar), and insoluble and soluble fiber. Total dietary fiber intake was higher with the FAT_LoCHO food relative to either other food but did not differ between the HiCHO and PROT_LoCHO foods. As anticipated based on the purpose-built nature of these foods, the intake of protein was highest on PROT_LoCHO, intake of carbohydrate was highest on HiCHO, and intake of fat was highest with FAT_LoCHO feeding. The differences in a given nutrient between the two foods that were not purposefully enriched in one of the three macronutrients led to intakes that, although statistically different, were not of a meaningfully large effect size; i.e., the intakes of carbohydrate in the two LoCHO foods were quite similar to each other relative to the HiCHO food. Only the FAT_LoCHO food was a significant source of the MCT-derived fats C8:0 and C10:0, as well as the n3 PUFA EPA and DHA. These fats were added to the FAT_LoCHO food in order to enhance ketogenesis (MCT) or to prevent ketogenic food-induced n3 PUFA depletion (EPA, DHA) [27].

### 3.3. Postabsorptive and Postprandial Responses of Biochemical Endpoints Relevant to Macronutrient Metabolism

Table 3 provides the 0-h (postabsorptive and fasted), 2-h (postprandial, fed), and differential values for biochemical endpoints related to macronutrient metabolism. At the fasting, postabsorptive 0-h point, the foods strongly modulated these endpoints. Albumin, BUN, bilirubin, BHB, TGs, total circulating fatty acids, and cholesterol all manifested a significant food effect. Additionally, postabsorptive albumin, BHB, TGs, and cholesterol were different between all three pairwise comparisons of foods. The increasing rank order of circulating levels for postabsorptive albumin paralleled the level of protein the dogs consumed from the respective foods (HiCHO, FAT_LoCHO, and PROT_LoCHO). Increased requirement for transport of circulating NEFAs as well as the possibility of dehydration were considered as factors driving differences in fasting-albumin across foods. This was prompted by the fact that NEFAs are transported in the circulation in part through their binding to albumin, and that reduction in blood volume may concentrate blood-borne analytes such as albumin. There was a weakly positive association between fat intake (g crude fat/(kg BW)^0.75^) and circulating albumin (*p* = 0.045) but adjusted r^2^ = 0.01, indicating that despite the statistical significance the association did not explain a meaningful amount of the variability in albumin. Levels of circulating NEFAs were also assessed for association with albumin. Similar to fat intake, there was a positive correlation between circulating NEFAs and albumin (*p* = 0.019) but there was little explanatory value (adjusted r^2^ = 0.03). Dehydration as a confounder of increased albumin was ruled out on the basis of the electrolyte patterns in the blood showing that the electrolyte levels in the dogs did not align with the pattern of albumin levels observed by food (Appendix A). Sodium, potassium, and chloride remained in normal ranges and were consistently decreased when dogs were fed the LoCHO foods relative to the HiCHO food. If dehydration were driving increased circulating albumin, the electrolyte levels should also have been concurrently elevated. Additionally, routine caretaker and veterinary examinations ruled out dehydration as a source of increased albumin. For comparison, the association between protein intake (g crude protein/(kg BW)^0.75^) and circulating albumin was statistically strong (*p* = 1.4 × 10^−11^) but still only explained less than 10% of the variation in levels of circulating albumin (adjusted r^2^ = 0.08). Circulating BUN was higher in the postabsorptive state for PROT_LoCHO than with the other foods but notably was not different between HiCHO and FAT_LoCHO despite FAT_LoCHO leading to higher albumin levels than HiCHO. Bilirubin, a marker of iron heme porphyrin protein intake and catabolism, was lowest in the FAT_LoCHO-fed dogs, although this only reached significance relative to when the dogs were fed the HiCHO food. The increasing rank order for postabsorptive circulating BHB, a marker of ketogenesis, paralleled the dietary fat intake levels (HiCHO, PROT_LoCHO, FAT_LoCHO). In contrast, the increasing rank order of TGs was not paralleled by the dietary intake of fat (FAT_LoCHO, PROT_LoCHO, HiCHO). For both BHB and TGs, there were minimal differences in magnitude between the two LoCHO foods and a much higher difference between both LoCHO foods and the HiCHO food. While there were minimal differences in NEFA levels in the postabsorptive state, both total fatty acids and cholesterol were markedly higher when dogs were fed the FAT_LoCHO food relative to either other food. Interestingly, the postabsorptive lipemic status was highest in dogs when they were fed the HiCHO food, and this reached significance relative to PROT_LoCHO. The FAT_LoCHO food had a higher ratio of fasting BHB/NEFAs, a proposed marker of hepatic ketogenesis [34], than either other food, while the PROT_LoCHO food was not different from the HiCHO food (Table 3).

At the 2-h postprandial point, every biochemical endpoint was significantly impacted by the food the dogs ate. Both LoCHO foods produced circulating postprandial glucose levels that were significantly lower than the HiCHO food but that were not different from each other. The BHB levels at 2 h presented as the following increasing rank order, with each pairwise comparison being significantly different from each other: HiCHO, PROT_LoCHO, FAT_LoCHO. The magnitude of difference between the two LoCHO foods at the postprandial point surpassed that observed at the 0-h postabsorptive point. All reported metrics of fat energy availability at the 2-h postprandial point were higher in the FAT_LoCHO-fed dogs relative to either other food, including NEFAs, TGs, total fatty acids, cholesterol, and lipemic status. Surprisingly, despite total fat intake being higher when dogs were fed the PROT_LoCHO food relative to the HiCHO food (Table 2), the postprandial levels of NEFAs and total fatty acids were significantly lower. Perhaps the most sizeable difference in postprandial biochemical metrics was with BUN. The postprandial BUN level stemming from the PROT_LoCHO food was numerically double as well as highly significant relative to that of either the FAT_LoCHO or HiCHO foods. While of much smaller magnitude, the postprandial level of BUN from FAT_LoCHO-fed dogs was higher than that of HiCHO-fed dogs.

When the difference between the 0-h postabsorptive point and the 2-h postprandial point was assessed for these biochemical metrics, all were significantly impacted by food, except for albumin, bilirubin, and cholesterol. Regarding postabsorptive ingestion of carbohydrate, the change (decrease) in glucose from the 0-h postabsorptive point to the 2-h postprandial point was significant for both LoCHO foods, and this decrease was also significantly different between both LoCHO foods versus the HiCHO food; the HiCHO food did not manifest a change. For postabsorptive indicators of fat absorption and metabolism, BHB significantly declined after consumption of the HiCHO food but increased after eating either LoCHO food, and the FAT_LoCHO food produced higher levels of BHB than the PROT_LoCHO food. The levels of TGs were increased after feeding according to the increasing rank order of HiCHO, PROT_LoCHO, FAT_LoCHO, which paralleled the levels of fat intake. Interestingly, the levels of NEFAs significantly decreased after consumption of either the HiCHO or PROT_LoCHO food but were unchanged after eating the FAT_LoCHO food. There was a significant increase in the BHB/NEFA ratio in the fasted–fed transition for all three foods; however, the increase was more than an order of magnitude larger with either LoCHO food than with the HiCHO food (Table 3). Further, the increase in BHB/NEFAs was not different between the two LoCHO foods and resulted in similar postprandial levels of BHB/NEFAs. With regard to postabsorptive markers of protein metabolism, the change in BUN paralleled the fasting BUN differences and increased in rank order as HiCHO, FAT_LoCHO, PROT_LoCHO, with significant differences in the responses to each of these foods. Creatinine, a marker of consumption of animal protein, was increased after consumption of each of the foods, but the increases were not different between LoCHO foods and each of the LoCHO foods was increased relative to the HiCHO food. In contrast to BUN or creatinine, the levels of bilirubin were decreased after feeding any of the study foods but the decreases for each food were not different from each other.

### 3.4. Postabsorptive and Postprandial Levels of Circulating Energy Forms

The postabsorptive, postprandial, and changes in levels of forms of circulating energy are shown in Table 4. There was a significant effect of food on the apparent total circulating energy and the percentage of total energy for every component of this composite metric in the postabsorptive phase. There was no difference in the postabsorptive level of apparent total circulating energy between the HiCHO and PROT_LoCHO food-fed dogs; however, the dogs fed the FAT_LoCHO food had significantly higher apparent total circulating energy than those fed either other food. There was also no difference between the HiCHO and PROT_LoCHO foods for the percentage of energy found as circulating glucose, but FAT_LoCHO feeding produced percentage energy levels of glucose that were significantly lower than for either other food. For the percentage of energy found as BHB, TGs (glycerol energy fraction only), and TGs (fat fraction only), there was no difference between the two LoCHO foods, but both of these foods had higher BHB and lower levels of both TG fractions than the HiCHO food. The postabsorptive levels of energy found as NEFAs were not different between the HiCHO food and either of the two LoCHO foods, but the two LoCHO foods were different from each other, with the PROT_LoCHO food having higher energy as NEFAs than the FAT_LoCHO food. The percentage of total energy found as adjusted total fatty acids (the sum of all analyzed fatty acids minus the fat fraction of TGs and NEFAs) was different between all foods, with the percentages found in increasing rank order of HiCHO, PROT_LoCHO, FAT_LoCHO. The bivariate correlation of NEFAs and TGs in the postabsorptive state was examined separately for dogs fed each of the three experimental foods. There was no significant correlation between fasting NEFAs and TGs when the dogs were fed either the HiCHO (R^2^ = 1.5 × 10^−4^; *p* = 0.944) or the PROT_LoCHO (R^2^ = 4.3 × 10^−3^; *p* = 0.739) foods. In contrast, however, fasting NEFA and TGs were positively correlated in dogs that ate the FAT_LoCHO food (R^2^ = 0.2; *p* = 0.009).

At the 2-h postprandial point, there was also a significant effect of food on the apparent total circulating energy and the percentage of total energy for each of its components (Table 4). The apparent total circulating energy was different between all foods and found in increasing rank order of PROT_LoCHO, HiCHO, FAT_LoCHO. The amount of energy circulating as glucose was not different between the HiCHO- and PROT_LoCHO-fed dogs but was lower than either of these two foods with FAT_LoCHO feeding. BHB and NEFAs were the only circulating energy components that differed between HiCHO and PROT_LoCHO in the postprandial phase. However, the percentage of total energy for every single component was different between the FAT_LoCHO food and either other food. BHB, both fractions of TGs (glycerol, fat), and NEFAs were higher with FAT_LoCHO than either other food, while adjusted total fatty acids were lower. In the fed state, the correlation between NEFAs and TGs was no longer present for the dogs fed FAT_LoCHO and remained absent for both the PROT_LoCHO and HiCHO foods.

The subtractive difference between the 0-h postabsorptive point and the 2-h postprandial point was assessed for circulating total energy and its components (Table 4). There was a significant effect of food type on changes in total circulating energy and every component of circulating energy besides glucose. When assessing the changes from the postabsorptive to postprandial phase after feeding within each food group, the only individual component that did not show a change from 0 to 2 h was glucose in the HiCHO group. Considering the magnitude of these changes, there was no difference in the increase in total circulating energy between feeding the HiCHO and PROT_LoCHO foods, but the FAT_LoCHO food led to an increase in energy that was greater than either of these two other foods. There was no difference between the HiCHO and PROT_LoCHO foods for the change in circulating glucose percentage of energy at the 2-h timepoint, but the FAT_LoCHO food led to a decrease in glucose that was significantly different to either other food. The changes in the components of circulating energy other than glucose were significantly different between all pairwise comparisons of the foods. Whereas the HiCHO food-fed dogs manifested a decrease in BHB at the 2-h time point, both LoCHO foods produced an increase in BHB, and the increase with the FAT_LoCHO food was greater than that for the PROT_LoCHO food. The increasing rank order of absolute magnitude of change in energy components when transitioning from the postabsorptive to postprandial phase after feeding for both fractions of TGs (glycerol and fat) and adjusted total fatty acids was HiCHO, PROT_LoCHO, FAT_LoCHO. In contrast, the increasing rank order for the absolute magnitude of changes in NEFAs was FAT_LoCHO, HiCHO, PROT_LoCHO; thus, the FAT_LoCHO food led to smaller decreases in NEFAs than either other food.

### 3.5. Postabsorptive and Postprandial Levels of Circulating Metabolite Classes Indicative of Macronutrient Metabolism

Untargeted metabolomics from serum samples at the postabsorptive and postprandial phases were used to assess levels of classes of metabolites related to macronutrient metabolism. Catabolic, anabolic, and signaling-type biochemicals were assessed both as multivariate and summed classes consisting of individual metabolites (Table 5). The log_2_ relative fold levels of all constituent members of these classes are available as Appendix A. In the 0-h postabsorptive state, every group of metabolites exhibited a significant effect of food when assessed as a multivariate class that considers the level of each individual constituent (MANOVA; *p* < 0.0001 for all classes). When assessed as a univariate summation of the individual constituents, all catabolic product classes except for α-hydroxy and ω-carboxy fatty acids showed a significant effect of food in a mixed model analysis. The only pairwise difference between foods for circulating postabsorptive amino acids was that PROT_LoCHO presented with lower levels than did FAT_LoCHO; HiCHO was intermediate and not different than either other food. In contrast, total circulating levels of the host and gut microbial putrefactive products of aromatic amino acid catabolism (indoles from tryptophan and, separately, phenols from phenylalanine or tyrosine) were lower in dogs fed either LoCHO food than when dogs consumed the HiCHO food (Appendix A). Although levels of indoles were not different between the LoCHO foods, phenols were lower in PROT_LoCHO-fed dogs than when those dogs were fed the FAT_LoCHO food. Thus, counterintuitively, the levels of circulating putrefactive catabolic products of amino acids were inversely ranked relative to the protein intake of the dogs. Of particular note, the renal toxin 3-indoxyl sulfate was significantly higher in dogs fed the HiCHO food relative to dogs fed either LoCHO food (Appendix A). β-hydroxy fatty acids and acylcarnitines were different between each pairwise comparison of the foods, and both of these catabolic lipid products had the increasing rank order of HiCHO, PROT_LoCHO, FAT_LoCHO. Postabsorptive serum mono- and diacylglycerides (MDAG) were not different between HiCHO and FAT_LoCHO, but both of these foods produced levels of MDAG that were significantly higher than the PROT_LoCHO food. When assessing the anabolic products of lipid metabolism, the rank order of fasting levels of glycerophosphocholines (GPCs) among the foods was the opposite of that for glycerophosphoethanolamines (GPEs). All of these pairwise comparisons were significantly different except that GPCs were not significantly different between the two LoCHO foods. Total sphingolipids (SPHINGs) were not different between HiCHO and PROT_LoCHO, but the FAT_LoCHO-fed dogs manifested higher levels of SPHINGs than either other food. Fatty acylcholines, acylethanolamides, and acyl amino acids were assessed as signaling-type lipid products. Interestingly, the FAT_LoCHO food produced levels of all of these classes of lipid-signaling mediators that were significantly higher than those of either other food. The only lipid-signaling class that differed between the HiCHO and PROT_LoCHO food groups was increased acylcholines in the latter food. The level of total circulating advanced glycation end products (AGEs) was significantly higher in dogs fed the HiCHO food relative to those fed either LoCHO food. This was despite the fact that the HiCHO food contained only ~1% total sugars; starch formed the overwhelming amount of carbohydrate in the food.

After feeding, similar to the observation for the postabsorptive state, every group of metabolites exhibited a significant effect of food when assessed as a multivariate class (MANOVA; *p* < 0.0001 for all classes). When assessed as a univariate summation of the individual constituents, all classes except for the GPC anabolic class of lipid products showed a significant effect of food (mixed model). Serum amino acids were significantly different between all groups and were found in the increasing rank order of HiCHO, FAT_LoCHO, PROT_LoCHO; this parallels the measured protein intakes and is the same rank order as was observed for BUN. Although both indoles and phenols increased after feeding in dogs fed PROT_LoCHO and phenols increased in dogs fed FAT_LoCHO, the level of total indole and phenol products of amino acid catabolism remained at levels far below those found when dogs were fed the HiCHO food. The FAT_LoCHO food resulted in higher levels of every catabolic lipid product class when compared to PROT_LoCHO (MDAGs, acylcarnitines, and α-hydroxy, β-hydroxy, and ω-carboxy fatty acids). The postprandial levels of these catabolic lipid products were also significantly higher in dogs fed the FAT_LoCHO food relative to those given the HiCHO food, except for MDAGs and α-hydroxy fatty acids, which were not different between these two foods. For anabolic lipid classes, while GPEs were different between each of the three foods and followed the increasing rank order of PROT_LoCHO, FAT_LoCHO, HiCHO, the GPCs were only different between the HiCHO and PROT_LoCHO foods. Similar to the postabsorptive state, SPHINGs were not different between HiCHO and PROT_LoCHO foods but were higher in the FAT_LoCHO-fed dogs than those in either other food group. Among the signaling-type lipids, each of the three classes of acylcholines, acylethanolamides, and acyl amino acids was different between each pairwise comparison of foods except that acylethanolamides were not different between the two LoCHO foods. Whereas the HiCHO food produced the highest levels of acylcholines, the FAT_LoCHO food produced the highest levels of acyl amino acids, and the PROT_LoCHO-fed dogs presented with the lowest levels of these two signaling lipid classes. Intriguingly, the level of total AGEs increased after a meal to a greater degree with the two LoCHO foods than it did with the HiCHO food, such that in the 2-h postprandial state the level of total AGEs was higher after feeding either LoCHO food than it was with the HiCHO food.

The changes from the 0-h postabsorptive point to the 2-h postprandial point for every single class of macronutrient-related metabolites were different when considered as multivariate composites (MANOVA; *p* < 0.0001 for all classes). When the classes were assessed as univariate sums of their constituent metabolites, all of the classes except for α-hydroxy fatty acids exhibited a food effect by linear mixed model. Between the FAT_LoCHO and both of the other foods, there was a striking difference in the change from fasting for acylcarnitines and β-hydroxy fatty acids; whereas FAT_LoCHO-fed dogs exhibited a large increase in both of these catabolic products of lipid metabolism, both HiCHO and PROT_LoCHO fed dogs manifested sizeable declines. The FAT_LoCHO-fed dogs had a significant increase in every single class of catabolic lipid metabolites. However, the FAT_LoCHO-fed dogs did not have increases in two of the three anabolic lipid classes; neither GPCs nor SPHINGs were changed by feeding. For signaling lipids, consumption of either LoCHO food significantly decreased the acylcholine and acylethanolamide classes after feeding, while the HiCHO food increased acylcholines and had no effect on acylethanolamide levels. In contrast to its effect to reduce acylcholine and acylethanolamide classes, the FAT_LoCHO food increased acyl amino acids. Neither other food affected the acyl amino acid class of lipid signaling intermediates.

### 3.6. Influence of Macronutrient Makeup and Fat-Type Content on Postabsorptive and Postprandial Levels of Individual Fatty Acids Derived from NEFAs and Complex Lipids

Individual fatty acids were quantitatively assessed from total circulating fatty acids originating from lipids, including NEFAs, TGs, phospholipids, and cholesteryl esters (Table 6). There was a wide-ranging impact of food on both postabsorptive and postprandial fatty acid levels, as 13/18 fatty acids in the fasted state, 18/18 fatty acids in the fed state, and 16/18 fatty acids assessed as the change from fasting to fed exhibited a significant effect of food. The FAT_LoCHO-fed dogs were the only dogs in whom C8:0 was detected and the only dogs in which C10:0 was detected in the postprandial state. The 10:0 fatty acid was only detected in the HiCHO or PROT_LoCHO foods in the postabsorptive state at an order of magnitude lower than that found in FAT_LoCHO-fed dogs. In the postabsorptive state, the HiCHO-fed dogs had levels of 12:0 that were higher than the FAT_LoCHO-fed dogs and levels of 14:0 that were higher than dogs fed either other food. In contrast, the FAT_LoCHO-fed dogs had higher levels of 16:0 than the HiCHO-fed dogs and higher levels of 18:0 than dogs fed either other food. Interestingly, only one fatty acid of 20 carbons or more at any degree of unsaturation changed from fasting to fed states with either LoCHO feeding (20:4n6), in spite of the FAT_LoCHO food containing supplemented 20:5n3 (EPA) and 22:6n3 (DHA). This response was different from the HiCHO-fed dogs, which had a fasting-to-fed state change in five/seven fatty acids with chain lengths of 20 or more. The FAT_LoCHO food changed the apparent desaturase and elongase enzyme activities in the fasted state. According to the ratio of 16:1/16:0, the FAT_LoCHO food had a lower stearoyl-CoA desaturase (SCD1) activity than the PROT_LoCHO food and lower SCD1 activity than either other food when the 18:1/18:0 ratio was assessed. Similarly, when fed the FAT_LoCHO food, dogs had reduced apparent d6 desaturase (18:3n6/18:2n6) relative to the PROT_LoCHO food and lower d5 desaturase (20:4n6/20:3n6) than either other food in the postabsorptive and postprandial states. Further, the FAT_LoCHO food relative to either other food had the highest apparent levels of elongation enzyme activity in the fasted state for both elongase Elovl-6 (18:0/16:0) and elongase Elovl-5 (20:3n6/18:3n6). These results also confirm the effect of the FAT_LoCHO food of changing the apparent desaturase and elongase enzyme activities in the fasted state, as reported previously for similar foods [27].

### 3.7. Differential Impact of Fat versus Protein Replacement of Carbohydrate on Circulating Satiety, Regulatory, and Counter-Regulatory Hormones in the Postabsorptive and Postprandial States

Table 7 shows that in the postabsorptive state, insulin, glucagon, the molar ratio of insulin/glucagon, glucagon-like peptide-1 (GLP-1), and pancreatic polypeptide (PP) did not exhibit an overall food effect, and further, there were no significant differences in the levels of these endpoints between any pairwise assessments of the foods. Gastric inhibitory peptide (GIP) was lower in the PROT_LoCHO-fed dogs, but this was the only pairwise difference for GIP in the fasted state. Pancreatic peptide YY (PYY) and leptin were not different in the comparison of HiCHO and FAT_LoCHO foods, but both of these foods produced higher levels of these hormones in the fasted state than did PROT_LoCHO. Postabsorptive ghrelin was higher with the FAT_LoCHO food than either other food, while the HiCHO and PROT_LoCHO food levels of ghrelin were not different from each other. Despite the aforementioned differences in the individual hormones leptin and ghrelin, the molar ratio of leptin/ghrelin was not different between the two LoCHO foods, and they both had significantly lower ratios of leptin/ghrelin than the HiCHO food produced in postabsorptive dogs.

Two hours after feeding there was a significant effect of food on 8/10 circulating hormones or ratios; only leptin and the leptin/ghrelin ratio did not exhibit an effect of food. The PROT_LoCHO feeding led to the highest levels of insulin and glucagon relative to either other food. The 2-h postabsorptive levels of insulin were not different between the HiCHO and the FAT_LoCHO foods, but glucagon was significantly higher after FAT_LoCHO feeding than with HiCHO feeding. The overall outcome was that the ratio of insulin/glucagon was lower with the FAT_LoCHO food than with the HiCHO food. The incretin hormones GIP and GLP-1 were both higher with FAT_LoCHO feeding than with either other food, and pancreatic polypeptide (PP) was lower with FAT_LoCHO than with either other food. Further, GIP was lower with PROT_LoCHO feeding in the fed state than with HiCHO feeding, but neither GLP-1 nor PP were different between these two foods. For PYY, the only pairwise difference was between the two LoCHO foods in which the FAT_LoCHO food produced higher levels of 2-h postprandial PYY than did PROT_LoCHO. Postprandial leptin did not differ between dogs fed any of the foods. Neither ghrelin nor the ratio of leptin/ghrelin were different between dogs fed the HiCHO and FAT_LoCHO foods, but levels of these endpoints were lower with these foods than with PROT_LoCHO feeding.

The assessment of the impact of the fasting-to-fed transition within individual foods showed that all three foods produced significant increases in insulin, glucagon, GIP, GLP-1, PP, and PYY. The insulin/glucagon ratio was increased by HiCHO feeding from 0 to 2 h, but this ratio was not changed by feeding either LoCHO food. Leptin only significantly changed from the fasted state with feeding of the PROT_LoCHO food. Ghrelin significantly decreased with either LoCHO food but was unchanged by feeding the HiCHO food. The outcome was that the leptin/ghrelin ratio was significantly increased after feeding either LoCHO food but unchanged with HiCHO feeding.

The comparison of the magnitude and direction of changes of the hormones when transitioning between the fasted to fed states across foods revealed that changes in the levels of several hormones exhibited an overall effect of food; only insulin, GIP, GLP-1, and PYY did not show differences in the magnitude of shifts across the foods. The postprandial increase in insulin was greater with PROT_LoCHO feeding than with FAT_LoCHO feeding but was not different in either other pairwise comparison of the change from fasted. The changes in glucagon were different between all pairwise comparisons of foods and were found in the following increasing rank order: HiCHO, FAT_LoCHO, PROT_LoCHO. The increase in the molar ratio of insulin/glucagon was significantly higher with HiCHO feeding than with both LoCHO foods; the ratios for the two LoCHO foods were not different. The increase in GIP was greatest in the FAT_LoCHO-fed dogs relative to either other food and the change with HiCHO was not different than with PROT_LoCHO feeding. The increase in GLP-1 after feeding was not different between the two LoCHO foods, but each of these foods led to increases in GLP-1 that were significantly greater than with HiCHO feeding; the increase with FAT_LoCHO feeding was nearly double that of HiCHO feeding. There were no differences in the responses of PYY to the three foods and the responses of PP were only different between the two LoCHO foods. The increase in leptin with PROT_LoCHO feeding was significantly higher than for either HiCHO or FAT_LoCHO, and the leptin responses to feeding the latter two foods were not different. Similarly, the decreases in ghrelin with either LoCHO food were not different, but these decreases were both significantly different to the lack of change occurring with HiCHO feeding. The overall impact to the molar ratio of leptin/ghrelin was that there were significant differences between the responses of this ratio to feeding between all three foods, with the HiCHO food not producing a change from the fasted state and the PROT_LoCHO food generating an increase that was greater than the increase observed with FAT_LoCHO feeding.

## 4. Discussion

### 4.1. Assessment of the Dynamics of Response to Macronutrient Intake

We previously reported the energy required to maintain weight and the postabsorptive characteristics of energy metabolism when dogs were fed foods that varied in macronutrient content [27]. However, there were no postprandial measurements in that trial and thus no ability to assess the change from fasted to fed states in dogs habituated to foods of differing macronutrient composition. There was also no assessment of the satiety, regulatory, or counter-regulatory hormone levels in response to the foods. Thus, the current intervention trial was conducted in dogs fed nearly identical foods to those in the preceding trial, with assessment of energy metabolism and hormone status at both the postabsorptive (0-h, approximately 23 h fasted) and 2-h postprandial states. The aim was to assess the responsiveness of dogs’ circulating energy and hormone levels as they transitioned from a fasted to fed state when consuming HiCHO, PROT_LoCHO, and FAT_LoCHO foods. Further, the current study also included the same 0-h postabsorptive time point as the previous study, allowing for the assessment of the degree to which the previously reported results are replicable.

#### 4.1.1. Macronutrient Composition Determines Postprandial Circulating Hormones and Energy Forms, Including Change from Postabsorptive State

In the preceding report, the increasing rank order of ME calories required to maintain body weight was HiCHO, PROT_LoCHO, FAT_LoCHO [27]. In the current study, the FAT_LoCHO food was again the highest in a ranked order of ME intakes which maintained weight, but the PROT_LoCHO food was lowest and the HiCHO food was intermediate. When considering historical results as well as the findings of the current trial, it would appear that macronutrient composition can have an impact on the caloric intake required to maintain BW. Although the effect size appears to be only moderate, the impact to BW maintenance across life stages bears consideration. Future work will be required to assess the degree to which body composition (e.g., lean or high fat mass) changes with these macronutrient profiles.

It has been documented that postprandial status does not prevent the assessment of common biochemical markers in dogs. While markers such as BUN and TGs increase during the transition from fasted to fed states, they stay within tolerable limits for analysis [16]. Further, postprandial lipemia also does not interfere with many typical biochemical endpoints [35], which is a consideration here as the current study fed a high-fat food, with a 2-h postprandial collection. The 2-h timepoint for postprandial assessment was selected based on existing literature that reports that this timepoint still presents sizable amounts of macronutrient-derived metabolites. The postabsorptive timepoint of 23 h post meal was chosen because the literature indicates that the postprandial state can exist for up to 12 h [4,8,24].

#### 4.1.2. Circulating Energy and Metabolic Hormones in the Fasted Postabsorptive State

The assessment of energy in the postabsorptive state showed that the FAT_LoCHO food led to the highest levels of total circulating energy; this energy metric was not different between the PROT_LoCHO and HiCHO foods. Further, the total circulating energy as percentage glucose was lower after FAT_LoCHO feeding than it was for either other food. This is despite the observations that glucose molarity, insulin, glucagon, and insulin/glucagon ratio did not differ by food type in the fasted state. With FAT_LoCHO feeding, energy was found to a greater degree in BHB and non-triglyceride, non-NEFA fatty acid fractions (e.g., complex lipids including acylcarnitines, cholesteryl esters, phospholipids, etc.) than with either other food, emphasizing the influence of this food on increasing fat catabolism relative to a high-carbohydrate or high-protein food. Both LoCHO foods had the same amount of energy found as triglyceride (fat or glycerol fractions), and these amounts were markedly lower than those found with the HiCHO food. Contrasting with the unique energy status induced by the FAT_LoCHO food, the groups fed the HiCHO and PROT_LoCHO foods had the same amount of total circulating energy, percentage of circulating energy found as glucose, and energy found as NEFAs.

One of the only examined hormones to differ between FAT_LoCHO and either of the other foods when fasted was ghrelin; this hormone was 58% higher with FAT_LoCHO feeding than with HiCHO feeding and 44% higher than with PROT_LoCHO feeding. Ghrelin has oppositional effects on fat mobilization versus oxidation in that it reduces adrenergic counter-regulatory-induced lipolysis and increases oxidation [36]. However, another report showed that high-fat feeding abrogates the ability of ghrelin to reduce lipolysis in a rat model [37]. The ketone body BHB is also directly anti-lipolytic by acting in a homeostatic negative feedback loop at adipocyte nicotinic acid receptors [38]. Further, BHB is indirectly anti-lipolytic by increasing insulin release from beta cells in the pancreas in the presence of adequate levels of glucose [39] or other central energy substrates [40]. In the current study, NEFAs were not increased with FAT_LoCHO feeding relative to HiCHO in the fasted state. However, the NEFA catabolic intermediates acylcarnitines and β-hydroxy fatty acids were increased with the FAT_LoCHO food relative to either other food. Thus, there is increased fat energy in the fasted state with the FAT_LoCHO food and increased levels of catabolic fatty acid metabolites, including acylcarnitines and β-hydroxy fatty acids. Taken together with the historical observations on the effect of ghrelin and the influence of high-fat feeding on ghrelin activity, it is concluded that the high-fat feeding with FAT_LoCHO in tandem with increased fasting ghrelin levels permitted lipolysis to continue unimpeded and also concurrently expedited fat catabolism. In support, the conclusion that FAT_LoCHO-fed dogs exhibited heightened fat metabolism in the fasted state relative to dogs fed either other food is evidenced by a significant correlation between NEFAs and TGs in dogs fed the FAT_LoCHO food but not in dogs fed the PROT_LoCHO or HiCHO foods. Further, there was a higher ratio of BHB/NEFAs, a metric of efficiency of ketogenesis [34], in the fasted state when dogs were fed the FAT_LoCHO food, but the ratio was not different between PROT_LoCHO and HiCHO.

Ghrelin is an orexigenic hormone that also impacts the physiology of multiple organ systems [41]. Ghrelin receptor activity is a therapeutic target for canines; capromorelin is a ghrelin receptor agonist used to increase food intake in dogs with poor appetence [42], such as those with cancer. However, in dogs, capromorelin has been shown to cause excessive postprandial glucose excursions [43], which may indicate that low-carbohydrate ketogenic dietary interventions that increase endogenous ghrelin in the fasted state may be a more benign method to improve appetence than pharmaceutical interventions. This could be a particularly important consideration for dogs undergoing therapy for cancer. In cancer, appetence can suffer, which impacts the dog’s ability to secure nutrition sufficient to stave off cachexia [44], and capromorelin is proposed to provide a benefit in this context [42]. However, many cancer cell types preferentially utilize glucose and gluconeogenic substrates for energy in a manner that facilitates their acquisition of molecules that support cellular proliferation [45]. Low-carbohydrate ketogenic foods are proposed to be conducive to cancer therapy by reducing the availability of glucose and increasing the percentage of energy available as BHB or fat [46], although a pleiotropic mechanism has been invoked [47], including a direct anti-tumorigenic effect of BHB [48]. Intriguingly, when dogs were habituated to the HiCHO food in the present study, they manifested no decrease in ghrelin two hours after a meal. In contrast, feeding either LoCHO food resulted in significant decreases in ghrelin. This may be indicative of the satiety-inducing capacity of high-protein [49] and high-fat [50] meals. Considered together, the decrease in postprandial glycemia, increase in an anabolic marker (albumin), increase in fasted ghrelin, and increase in BHB indicate that the replacement of dietary carbohydrate with fat may offer therapeutic advantages beyond a drug such as capromorelin.

The relative levels of hormones can provide information on metabolic predilections as well as predisposition to morbidity, as noted above for the insulin/glucagon ratio. In humans, a higher leptin/ghrelin ratio is associated with overweight and obesity [51], high body mass index and diabetes [52], regaining of weight after weight loss in overweight/obese patients [53], and intrapancreatic fat deposition (in the fasted state) [54]. In the current study, consumption of either LoCHO food produced lower leptin/ghrelin ratios in the fasted state relative to the HiCHO food. However, this lower ratio originated in different ways for each of the LoCHO foods. With the PROT_HiCHO food, the ratio was lower due to leptin being reduced relative to either other food. In contrast, with FAT_LoCHO feeding the ratio was reduced due to increased ghrelin relative to either other food. If indeed the ratio of leptin/ghrelin per se is a mechanistic determinant of health and disease, then it would appear that there are two routes to achieve this end with LoCHO ketogenic foods: substitution of either protein or fat for carbohydrate energy.

#### 4.1.3. Circulating Energy during the Transition to the Postprandial State

The total circulating energy increased with all foods two hours after feeding. However, the increase in circulating energy was greater with the FAT_LoCHO food compared to either other food (183% and 152% higher than the HiCHO and PROT_LoCHO foods, respectively). This demonstrates that the increased fasting energy status found with FAT_LoCHO is propagated into the postabsorptive state. When fed, the percentage of energy found as glucose was not different from the fasted state for the HiCHO-fed dogs, and only the FAT_LoCHO food produced a decrease that was significantly different than HiCHO. In contrast, the delta values for 2 h–0 h glucose energy were not significantly different for PROT_LoCHO vs HiCHO. In terms of percentage total energy, there was a reduction in NEFAs for all foods. However, the decrease in NEFAs during the postprandial state was less with FAT_LoCHO feeding than the decrease with either other food (70% and 78% less NEFA reduction than HiCHO and PROT_LoCHO, respectively). Further, the FAT_LoCHO food was the only one of the three foods that did not lead to a decrease in the molar amount of NEFAs. This is surprising because a decrease in NEFAs is a canonical response to foods that result in an increase in insulin [55], and insulin was significantly elevated at 2 h post-meal with all three foods. In this light, it is not surprising that the greatest decrease in NEFAs with food consumption was when dogs were fed the PROT_LoCHO food, as this food produced an accompanying increase in insulin that was greater than that of the FAT_LoCHO food. The percentage of total energy and molar amount of BHB was increased from the fasted to fed state by feeding either LoCHO food, while they were decreased by feeding the HiCHO food. A decrease in BHB typically occurs with a meal that invokes a rise in insulin levels (as the LoCHO foods did) [56]. In contrast to this expectation, the increase in BHB with the LoCHO foods was accompanied by increases in insulin. The increase in BHB with FAT_LoCHO feeding was greater than with the PROT_LoCHO food; a contributing factor to this may be the inclusion of ketogenic medium-chain TGs into the FAT_LoCHO food. Alternatively, the PROT_LoCHO food led to an increase in postprandial insulin that was significantly greater than the fasted to fed increase that occurred with the FAT_LoCHO food. Perhaps the increased insulin response to the PROT_LoCHO food relative to the FAT_LoCHO food dampened the increase in BHB that occurred with the PROT-LoCHO feeding.

Between the fasted state and 2 h after eating, all foods produced an increase in both insulin and glucagon. However, the change in insulin/glucagon ratio significantly increased from fasting only for the HiCHO-fed dogs, but not for dogs fed either of the LoCHO foods. In humans, the postprandial ratio of insulin/glucagon is an inverse indicator of the degree to which endogenous glucose production is required to maintain homeostatic control of circulating glucose energy; the higher the insulin/glucagon ratio, the less endogenous glucose production occurs [57]. In the present study, although there were no increases from fasting in the insulin/glucagon ratio with either LoCHO food, there was a significantly higher increase in both insulin and glucagon with PROT_LoCHO feeding than with FAT_LoCHO. It may be that not only the absolute level of insulin, but also the ratio of insulin/glucagon, are important determinants of BHB response to a meal in dogs. Here, the insulin/glucagon ratio significantly increased from fasted–fed states with HiCHO feeding, which is expected to decrease lipolysis and ketogenesis. An insulin/glucagon ratio that is not increased with feeding, but rather after feeding, and maintained at levels found in the fasted state may allow the increases in BHB that were observed with both LoCHO foods.

Supplementation in humans with 20:5n3 fatty acid (EPA) has been shown to reduce fasting BHB and alter the postprandial response of BHB in humans, both in an age-dependent manner [11]. In contrast to the findings from that report, in the current study the fatty acid 20:5n3 was added to the FAT_LoCHO food and yet the FAT_LoCHO-fed dogs still showed the highest levels of postprandial BHB. Since the FAT_LoCHO food was the only food to contain MCFA (C8:0, C10:0), it is not surprising that it was the only food to produce increased postprandial levels; they were undetectable after feeding the other foods, which had not been supplemented with them. The MCTs are more readily digested by lipases in vitro [58] and ex vivo [59] while an in vitro hepatocyte model demonstrated that diacylglyceride synthase catalyzes the incorporation of C8:0 and C:10 into TGs, but these MCFAs are not incorporated into phospholipids [60]. The C8:0 and C10:0 MCFAs are primarily absorbed through the portal vein, with a relatively minimal amount absorbed into chylomicrons in lymphatic tissue in a rodent model [61], while a human study showed that while relatively little TG formed in chylomicrons with MCT-feeding relative to LCT-feeding, some MCFA-containing TGs were found in chylomicrons [62]. Rodent models have demonstrated that MCFAs are primarily catabolized to energy and that high-MCT-feeding results in less adipose deposition [63,64]. A study in humans showed that decreasing chain lengths in the range of C:12 to C:18 resulted in increasing oxidation [65], and substituting MCT for longer-chain TGs in the food leads to greater weight loss in humans [66]. A study in formula-fed infants showed apparent decreases in levels of the incorporation of dietary C8:0 and C10:0 (true MCFAs) than C12:0 (an intermediate FA) and longer fatty acids, such that adipose C8:0 and C:10 were much less associated with diet than were C:12 and higher chain-length fats [67,68]. Taken altogether, it would appear that relative to longer-chain fatty acids, C8:0 and C10:0 are more readily digested, absorbed, and oxidized. When these MCFAs are stored as TGs, they are then labile with respect to mobilization from adipose tissue into circulation (where they are presumably also more readily oxidized than LCFAs). It is thus interesting to note that there were detectable circulating levels of 8:0 and relatively high levels of 10:0 in the fasted state with the FAT_LoCHO food in the present study. The presence of circulating C:8 and C:10 in the fasted state may reflect a labile pool of MCFAs arising from postprandial storage in TG form in adipose tissue.

The hormones GIP and GLP-1 are produced by K and L cells, respectively, in the canine proximal small intestine (duodenum and jejunum) [69] and have pleiotropic physiological effects related to meal metabolism [70,71]. Both of these hormones are insulinotropic incretins [72,73], whose levels increase in response to ingestion of glucose, fat, and amino acids [25,74,75,76]. In dogs, GLP-1 may act in an indirect manner to increase insulin in the postprandial state [77,78]. The release of GIP and GLP-1 are interactive; in dogs, GIP appears to stimulate the release of GLP-1 [79], perhaps through a paracrine effect [69], and a GLP-1 receptor agonist increases levels of GIP [43]. The mechanisms of macronutrient stimulation of GIP-release may differ between glucose and fat [80]. It appears that diglycerides may be the stimulating agent for GIP- and GLP-1-release; fatty acids per se do not increase GIP [76], feeding of diglyceride increases incretins in dogs [81], and inhibition of the conversion of diglycerides to TGs increases GLP-1 in a genetically modified mouse model [82]. Further, in dogs, certain proteogenic amino acids decrease the GIP-releasing effect of fat, and the quantitative release of GIP is greater from dietary fat than glucose [76]. In the current study, despite there being negligible differences in fasting GIP and GLP-1 resulting from consumption of the three foods, the FAT-LoCHO food resulted in the highest postprandial GIP and GLP-1 levels relative to either other food. This is intriguing in light of the insulin results. Whereas these hormones are canonically considered to be incretins, the PROT_LoCHO food led to a larger increase in insulin than did FAT_LoCHO, but concurrently led to smaller increases in GIP and GLP-1. Further, there were no differences in the rises in insulin after a meal between HiCHO and FAT_LoCHO, but the FAT_LoCHO food led to significantly larger increases in GIP and GLP-1 than did HiCHO. Since the action of GLP-1 is an emerging target of pharmaceutical therapy in dogs [83], it would appear that the replacement of carbohydrate with fat rather than protein may provide a metabolic advantage.

### 4.2. Metabolism of Protein and Nitrogen Disposition in the Postabsorptive and Postprandial States

When considering the fasting and fed states and dynamics of the fasting-to-fed transition of nitrogen, a longitudinal analysis in dogs has shown that BUN may be a better marker of fasting body protein and postprandial dietary protein catabolism than creatinine [84]. The presence of a diurnal circadian pattern for creatinine exists in dogs and is confounded by food intake [85]. The postprandial status of dietary protein-derived nitrogen metabolites BUN and creatinine after a single meal of various protein types has been reported for dogs and showed that BUN had largely returned to baseline by 24 h post meal [21]. When considering the source of nitrogen-containing metabolites (muscle tissue for creatinine and proteolysis of amino acids for BUN), it would appear that BUN is less confounded than creatinine by ingredient type (e.g., animal versus plant proteins) and the method of cooking (e.g., extrusion) since creatinine is formed to variable amounts from creatine during the cooking of meat-based ingredients. Another report tracked fasting levels of BUN with habitual consumption of foods varying in carbohydrate, protein, and fat levels [86]. The authors reported that increased protein intake, but not the carbohydrate/fat ratio in the food, was a determinant of fasting BUN, with higher protein intake leading to higher BUN levels. In that study, the consumption of a food that contained 46–48% ME as protein energy increased fasting BUN above that observed for dogs consuming 23% or 36% ME as protein energy. In our previous report [27], we noted that the habituation of dogs to foods differing in protein levels led to a level of fasting BUN that was higher when dogs consumed 53% ME protein energy than that observed when they consumed foods that were 25–27% ME protein energy. In the current trial, the same observation was made. In this current trial, feeding 53% ME as protein energy led to a fasting BUN that was higher than with either a 25% or a 27% protein ME food. Considering all the aforementioned studies alongside the current one, a protein ME percentage of at least 45% increases fasting BUN relative to foods with protein ME of less than 35%. Since the BUN present at nearly 24 h post-meal is not derived from the preceding meal’s protein content but rather from endogenous catabolism of body proteins, we conclude that feeding a protein ME of about 50% to dogs appears to promote body protein catabolism in the fasting state.

In the present study, the increase in BUN at 2 h post-meal followed the level of protein intake (HiCHO, FAT_LoCHO, PROT_LoCHO). Postprandial BUN in dogs stems in a large part from hepatic gluconeogenic catabolism of circulating amino acids derived from dietary proteins, a process driven by the presence of the counter-regulatory hormone glucagon [87]. While the insulin/glucagon ratio did not differ in the postprandial state between dogs fed either LoCHO food, it could be that in addition to the higher protein intake, a higher absolute amount of circulating postprandial glucagon partially underpinned the increased BUN observed with PROT_LoCHO feeding.

Notably, the fasting albumin was increased in both LoCHO foods relative to the HiCHO food, but BUN was only increased relative to the HiCHO-fed dogs in the PROT_LoCHO-fed dogs. It is concluded that albumin protein synthesis increases along a spectrum of intakes from 7 g/(kg BW)^0.75^ found in the HiCHO food to 14 g/(kg BW)^0.75^ found in the PROT_LoCHO food, but the catabolism of body proteins to increase circulating BUN does not increase until protein intakes are higher than at least the 9 g/(kg BW)^0.75^ found in the FAT_LoCHO food. The level of protein in all of the foods was greater than the minimum level allowed in dog foods by the guiding regulatory body in the USA [88]. Also, the level of protein intake on a metabolic BW basis was greater than the recommended intake level for dogs [89]. Thus, the dogs eating even the lowest amount of protein (when consuming HiCHO) were eating 2.16 times the recommended level (9 g/(kg BW)^0.75^). According to current understanding, the recommended protein should be more than sufficient to maximize albumin levels. However, in contrast, the current data indicate that even higher levels of protein intake (FAT_LoCHO = 2.61 times and PROT_LoCHO = 4.26 times the NRC recommendation) may further increase albumin. Factors other than protein intake that were considered (fat intake and subsequent NEFA transport, as well as the possibility of dehydration) did not explain a meaningful amount of the variation in circulating albumin. Taken together, it would appear that factors other than protein and fat intake, NEFA levels, and dehydration are needed to explain why albumin levels differed across diets, but in both the preceding [27] and current reports, levels of albumin paralleled protein intake. It thus appears that protein intake at levels far greater than regulatory recommendations increase albumin relative to recommended levels.

The PROT_LoCHO food led to fasting levels of total circulating amino acids that were significantly lower than the levels found for the FAT_LoCHO food and nominally lower than found with HiCHO feeding (*p* = 0.102). Although summed classes were not considered in the preceding trial report [27], when the reported amino acids from that trial are assessed, PROT_LoCHO feeding led to the lowest levels of fasting circulating amino acids, which emphasizes the robustness of this observation. This phenomenon of reduced carbohydrate intake leading to lower levels of non-essential circulating amino acids in the postabsorptive state was also reported by another group [90], but a separate group reported minimal influence of carbohydrate at 16 weeks feeding and no effect at 32 weeks feeding [86]. The current study showed that in the fasted state, the dogs fed the LoCHO foods had reduced alanine, glutamate, and glutamine, each of which is a preferred source of energy and central metabolic products for multiple cancer subtypes [91,92,93]. With consistent observations that a low-carbohydrate food can decrease circulating non-essential amino acids, these foods might be considered for subsequent investigations to reduce the availability of preferred fuel sources for tumors.

One non-enzymatic route of amino acid disposal is the formation of AGE adducts. These AGE products of amino acid degradation are produced by the reaction of free or protein-bound amino acids with circulating sugars. Humans can consume significant amounts of their dietary carbohydrate intake as sugar (primarily sucrose). However, sugar is not prominent in pet foods and the HiCHO food employed contained only ~1% of the diet as sugar, with the overwhelming majority of dietary carbohydrate present as starch. Thus, the current report is notable because it shows that even when sugar intake is restricted, high carbohydrate intake in the form of starch can lead to significant amino acid glycation. Nearly every AGE observed, as well as total circulating AGE, was increased by the HiCHO food, including 1-carboxyethylvaline, which is reported to be associated with poor glycemic control in humans [94]. While there was no difference in fasting blood glucose across the foods, the 2-h postprandial blood glucose level was significantly increased when dogs were fed the HiCHO food relative to the LoCHO foods. It may be that fasted–fed transitional glycemia drives the overall increase in fasted levels of AGE, or perhaps that the snapshot of fasting blood glucose reported here does not provide reliable information on the predisposition toward AGE accumulation in dogs. Future investigations will assess glycated hemoglobin (HbA1C) as a longer-term marker of glycemia in response to feeding foods of differing macronutrient makeups.

Counterintuitively, levels of amino acid catabolic products, including gut microbial putrefactive postbiotics (indoles and phenols), were found to be lower when dogs were fed either LoCHO food relative to when they were fed the HiCHO food, despite the dogs’ consumption of more protein with the LoCHO foods. This result indicates that increased protein intake does not necessarily lead to increased circulating putrefactive uremic solutes and that gut microbial putrefactive processes may be stimulated by high carbohydrate intake. The protein:fiber ratio of a diet has been reported to be positively associated with circulating levels of 3-indoxyl sulfate in people with chronic kidney disease [95]; putrefactive microbial metabolites were increased with the HiCHO food in the current report. Further, addition of fiber [96] or prebiotic resistant starch (a de facto fiber) [97] to canine foods shifts the balance of microbial metabolism towards saccharolytic and away from putrefactive processes. In addition, the protein:starch ratio of feline foods is positively associated with production of microbial putrefactive markers even when the high protein:starch food delivers less protein to the colon due to increased protein digestibility [98]. However, the protein:fiber ratio (or more generally, protein:carbohydrate) cannot be the sole driver of microbial indole and phenol accumulation in the blood for dogs. In the current report, the HiCHO food, which increased fasting indoles and phenols, had the same protein:total fiber, protein:soluble fiber, and protein:insoluble fiber ratios as the FAT_LoCHO food as well as ratios for these three protein:fiber metrics that were much lower than in the PROT_LoCHO food. Furthermore, the HiCHO food also had a protein:starch (and thus protein:resistant starch) level that was much lower than either of the LoCHO foods. Thus, it would be expected that dogs consuming the HiCHO food would manifest the lowest levels of circulating putrefactive postbiotics. However, the current report supports a new perspective that dietary restriction of digestible carbohydrate can decrease circulating uremic indoles and phenols. Recent research in animal models and humans suggests that a ketogenic diet is a powerful modulator of the hindgut microbiome [99,100,101] and can decrease specific populations of microbes with particular metabolic functionality to improve inflammatory disease [102]. At least one study has shown that fecal putrefactive postbiotics (e.g., polyamines) are decreased by a ketogenic diet in a rodent model of Parkinson’s disease [103]. However, the quantitative and qualitative degree to which food that is exceptionally low in digestible carbohydrate impacts the canine circulating microbial-derived metabolome requires more characterization. To contextualize these counterintuitive results, one hypothesis is that resistant starch present in the HiCHO food could be increasing gut microbial biomass with a resultant increase in gut microbial foraging for nitrogen from unabsorbed digesta [104,105].

While a broad swath of both indole and phenol uremic solutes were increased by high carbohydrate feeding in the current study, a couple of these bear particular consideration. Firstly, 3-indoxyl sulfate is a renal toxin [106] that was increased with HiCHO feeding; this shows that the restriction of dietary digestible carbohydrate may have potential renal benefits in dogs. It has been proposed that a ketogenic diet may protect against renal disease in humans [107]. The observation here that a uremic, putrefactive product of the gut microbiome is decreased by both forms of LoCHO food despite these foods providing higher protein intakes is additional evidence that ketogenic foods may protect against renal disease through uncharacterized mechanisms. Secondly, levels of the gut microbial putrefactive postbiotic 4-ethylphenyl sulfate (4-EPS) and its congener 4-vinylphenyl sulfate (4-VPS) were both significantly increased by feeding the HiCHO food relative to the LoCHO foods. 4-EPS and its congeners are mechanistically linked to autism and anxiety in humans and in rodent models [108,109,110]. Ketogenic diets have been proposed to safely and effectively aid in the management of autism and anxiety [111,112]. The observation in the current report that the two diverse LoCHO foods (replacing carbohydrate with either protein or fat) decrease both 4-EPS and 4-VPS relative to a high-carbohydrate food provides additional justification for the investigation of low-carbohydrate foods in the management of canine fear, anxiety, and stress.

## 5. Conclusions

In the transition from the fed to fasted states, there was an increase in circulating energy after feeding a food in which fat replaced carbohydrate at a level twice that in the high-carbohydrate food or a food where protein replaced carbohydrate. Furthermore, previously reported findings were repeated, including the observation that in the fasted state there was an increase in circulating energy and a shift in the types of energy available when dogs were fed foods in which fat replaced dietary carbohydrate. The current findings of postprandial changes in hormones provide information on how macronutrients can influence dietary energy processing, postprandial metabolism, and satiety. Further, observations on the metabolism of amino acids as well as their putrefactive indole and phenol products indicate promise for low-carbohydrate foods to aid in the management of cancer, renal disease, and anxious behaviors. A caveat is that all subjects were healthy, and future work will be required to determine the degree to which modulating macronutrient levels might benefit diverse conditions including metabolic disease, cancer, inflammatory bowel disease, renal failure, or behavioral abnormalities.

## Figures and Tables

**Table 1 metabolites-14-00373-t001:** Analyzed nutrient content of experimental foods.

Food Component	HiCHO	PROT_LoCHO	FAT_LoCHO
Ketogenic ratio	0.46	1.08	1.65
Metabolizable energy, kcal/kg	3728.3	4026.9	4484.4
Protein, % kcal	25.4	51.5	29.6
Fat, % kcal	35.3	42.6	66.2
Carbohydrate, % kcal	39.3	5.9	4.2
Protein	23.7	51.9	33.2
Fat	14.6	19.1	33.0
Carbohydrate	36.6	5.9	4.7
Starch	35.5	5.4	3.9
Sugars	1.1	0.5	0.8
Total dietary fiber	10.6	11.2	14.7
Insoluble fiber	8.5	10.2	12.1
Soluble fiber	2.1	1	2.6
SFAs	3.6	5.25	14.29
MUFAs	5.17	7.38	10.52
n3 PUFAs	0.45	0.81	2
n6 PUFAs	3.61	3.74	4.23
n6/n3 Ratio	8.02	4.62	2.12
C8:0	<0.02	<0.02	4.04
C10:0	<0.02	<0.02	3.11
C12:0	<0.02	<0.02	<0.02
C14:0	0.06	0.09	0.2
C16:0	2.76	3.85	5.36
C16:1	0.59	0.96	1.42
C18:0	0.7	1.2	1.45
C18:1	4.48	6.28	8.89
C18:2n6	3.5	3.35	3.89
C18:3n6	<0.02	0.04	0.05
C18:3n3	0.43	0.75	1.6
C20:3n6	<0.02	0.05	0.04
C20:4n6 (ARA)	0.05	0.21	0.17
C20:5n3 (EPA)	<0.02	<0.02	0.18
C22:6n3 (DHA)	<0.02	<0.02	0.13

Values are presented as g/100 g food as fed, unless otherwise noted. Sugars are a sum of individual analytical values for fructose, glucose, lactose, maltose, and sucrose. Total dietary fiber is a sum of the analytical values for insoluble and soluble fiber. ARA—arachidonic acid; DHA—docosahexaenoic acid; EPA—eicosapentaenoic acid; MUFAs—monounsaturated fatty acids; PUFAs—polyunsaturated fatty acids; SFAs—saturated fatty acids.

**Table 2 metabolites-14-00373-t002:** Intake of metabolizable energy and dietary nutrients.

Nutrient Intake	HiCHO	PROT_LoCHO	FAT_LoCHO	Mixed Model *p*
Metabolizable energy	111.52 ± 2.67 ^a^	108.45 ± 3.33 ^b^	115.74 ± 4.35 ^a^	<0.001
Crude protein	7.09 ± 0.17 ^c^	13.97 ± 0.43 ^a^	8.57 ± 0.32 ^b^	<0.001
Crude fat	4.38 ± 0.10 ^c^	5.13 ± 0.16 ^b^	8.52 ± 0.32 ^a^	<0.001
Carbohydrate	10.95 ± 0.26 ^a^	1.60 ± 0.05 ^b^	1.21 ± 0.05 ^c^	<0.001
Starch	10.62 ± 0.25 ^a^	1.45 ± 0.04 ^b^	1.01 ± 0.04 ^c^	<0.001
Sugars	0.33 ± 0.01 ^a^	0.14 ± 0.00 ^c^	0.20 ± 0.01 ^b^	<0.001
Total dietary fiber	3.17 ± 0.08 ^b^	3.02 ± 0.09 ^b^	3.79 ± 0.14 ^a^	<0.001
Insoluble fiber	2.54 ± 0.06 ^c^	2.75 ± 0.08 ^b^	3.12 ± 0.12 ^a^	<0.001
Soluble fiber	0.63 ± 0.02 ^b^	0.27 ± 0.01 ^c^	0.67 ± 0.03 ^a^	<0.001
SFAs	1.08 ± 0.03 ^c^	1.41 ± 0.04 ^b^	3.69 ± 0.14 ^a^	<0.001
MUFAs	1.55 ± 0.04 ^c^	1.99 ± 0.06 ^b^	2.72 ± 0.10 ^a^	<0.001
n3 PUFAs	0.13 ± 0.00 ^c^	0.22 ± 0.01 ^b^	0.52 ± 0.02 ^a^	<0.001
n6 PUFAs	1.08 ± 0.03 ^a^	1.01 ± 0.03 ^b^	1.09 ± 0.04 ^a,b^	<0.001
C8:0	<0.007 ^b^	<0.006 ^c^	1.04 ± 0.04 ^a^	NA
C10:0	<0.007 ^b^	<0.006 ^c^	0.80 ± 0.03 ^a^	NA
C12:0	<0.007 ^a^	<0.006 ^b^	<0.006 ^c^	NA
C14:0	0.02 ± 0.00 ^c^	0.02 ± 0.00 ^b^	0.05 ± 0.00 ^a^	<0.001
C16:0	0.83 ± 0.02 ^c^	1.04 ± 0.03 ^b^	1.38 ± 0.05 ^a^	<0.001
C16:1	0.18 ± 0.00 ^c^	0.26 ± 0.01 ^b^	0.37 ± 0.01 ^a^	<0.001
C18:0	0.21 ± 0.01 ^c^	0.32 ± 0.01 ^b^	0.37 ± 0.01 ^a^	<0.001
C18:1	1.34 ± 0.03 ^c^	1.69 ± 0.05 ^b^	2.29 ± 0.09 ^a^	<0.001
C18:2n6	1.05 ± 0.03 ^a^	0.90 ± 0.03 ^b^	1.00 ± 0.04 ^a^	<0.001
C18:3n3	0.13 ± 0.00 ^c^	0.20 ± 0.01 ^b^	0.41 ± 0.02 ^a^	<0.001
C18:3n6	<0.007 ^c^	0.01 ± 0.00 ^b^	0.01 ± 0.00 ^a^	NA
C20:3n6	<0.007 ^c^	0.01 ± 0.00 ^a^	0.01 ± 0.00 ^b^	NA
C20:4n6 (ARA)	0.02 ± 0.00 ^c^	0.06 ± 0.00 ^a^	0.04 ± 0.00 ^b^	<0.001
C20:5n3 (EPA)	<0.007 ^b^	<0.006 ^c^	0.05 ± 0.00 ^a^	NA
C22:6n3 (DHA)	<0.007 ^b^	<0.006 ^c^	0.03 ± 0.00 ^a^	NA

Values other than metabolizable energy are presented as mean ± standard error of daily g intake/(kg BW^0.75^). Metabolizable energy was calculated by the Atwater equation using the sum of total sugars and starch for carbohydrate, and values are presented as mean ± standard error of daily kcal intake/(kg BW^0.75^). Sugars is a sum of individual analytical values for fructose, glucose, lactose, maltose, and sucrose. Carbohydrate is a sum of sugars and starch. Total dietary fiber is a sum of the analytical values for insoluble and soluble fiber. Superscript letters differing within a row across foods denote means that are significantly different at *p* < 0.05 by Wilcoxon signed-rank test. NA is used to designate those comparisons in which a nutrient was below the limit of quantitation. ARA—arachidonic acid; BW—body weight; DHA—docosahexaenoic acid; EPA—eicosapentaenoic acid; MUFAs—monounsaturated fatty acids; NA—not applicable; PUFAs—polyunsaturated fatty acids; SFAs—saturated fatty acids.

**Table 3 metabolites-14-00373-t003:** Effects of test foods on postabsorptive and postprandial responses of macronutrient metabolism-related biochemical endpoints.

	0 h			2 h			2 h−0 h			Mixed Model *p* Value (Overall Food Effect)		
Analyte	HiCHO	PROT_LoCHO	FAT_LoCHO	HiCHO	PROT_LoCHO	FAT_LoCHO	HiCHO	PROT_LoCHO	FAT_LoCHO	0 h	2 h	2 h−0 h
Glucose (mM)	5.06	5.01	5.04	5.17 ^a^	4.84 ^b^	4.65 ^b^	0.11 ^a^	−0.18 ^b^*	−0.25 ^b^*	0.804	<0.001	0.004
Albumin (g/dL)	3.47 ^c^	3.65 ^a^	3.57 ^b^	3.51 ^b^	3.69 ^a^	3.56 ^b^	0.04 ^b^	0.04 ^b^*	0.09 ^a^	<0.001	<0.001	0.827
BUN (mg/dL)	13.44 ^b^	18.42 ^a^	13.52 ^b^	16.28 ^c^	31.73 ^a^	17.41 ^b^	2.84 ^c^*	13.31 ^a^*	4.26 ^b^*	<0.001	<0.001	<0.001
Bilirubin, total (mg/dL)	0.08 ^a^	0.07 ^a,b^	0.05 ^b^	0.04 ^a^	0.02 ^b^	0.01 ^b^	−0.04 *	−0.04 *	−0.04 *	0.005	0.001	0.885
Creatinine (mg/dL)	0.69	0.72	0.73	0.75 ^b^	0.83 ^a^	0.85 ^a^	0.06 ^b^*	0.11 ^a^*	0.14 ^a^*	0.124	<0.001	0.001
β-hydroxybutyrate (mM)	0.08 ^c^	0.11 ^b^	0.13 ^a^	0.06 ^c^	0.14 ^b^	0.26 ^a^	−0.02 ^c^*	0.03 ^b^*	0.13 ^a^*	<0.001	<0.001	<0.001
NEFAs (mM)	0.77 ^a,b^	0.83 ^a^	0.73 ^b^	0.44 ^b^	0.36 ^c^	0.64 ^a^	−0.33 ^b^*	−0.46 ^c^*	−0.10 ^a^	0.247	<0.001	<0.001
β-hydroxybutyrate/NEFAs	127.05 ^b^	129.91 ^b^	187.58 ^a^	153.91 ^b^	418.26 ^a^	433.12 ^a^	26.86 ^b^*	288.35 ^a^*	245.54 ^a^*	<0.001	<0.001	<0.001
Triglycerides (mM)	0.75 ^a^	0.42 ^c^	0.47 ^b^	1.77 ^b^	1.90 ^b^	3.27 ^a^	1.02 ^c^*	1.48 ^b^*	2.81 ^a^*	<0.001	<0.001	<0.001
Total fatty acids (mM)	9.46 ^b^	9.28 ^b^	10.67 ^a^	9.79 ^b^	9.43 ^c^	11.62 ^a^	0.33 ^b^*	0.15 ^b^	0.96 ^a^*	<0.001	<0.001	<0.001
Cholesterol (mM)	5.15 ^c^	5.41 ^b^	6.47 ^a^	5.12 ^c^	5.40 ^b^	6.37 ^a^	−0.03	−0.01	0.08 *	<0.001	<0.001	0.750
Lipemic index (mg/dL)	7.31 ^a^	6.28 ^b^	6.54 ^a,b^	62.86 ^b^	79.31 ^b^	128.06 ^a^	55.56 ^b^*	73.03 ^b^*	121.69 ^a^*	0.133	<0.001	<0.001

Superscript letters differing within a row across foods for a timepoint denote mean values that are significantly different at *p* < 0.05 by Wilcoxon signed-rank test. Asterisks indicate significant differences between the 2 h and 0 h timepoints within an individual test food by Wilcoxon signed-rank test. Full data, including standard errors, are available in Appendix A. BUN—blood urea nitrogen; NEFAs—non-esterified fatty acids.

**Table 4 metabolites-14-00373-t004:** Effects of test foods on postabsorptive and postprandial levels of circulating energy forms.

	0 h			2 h			2 h−0 h			Mixed Model *p* Value (Overall Food Effect)		
Analyte	HiCHO	PROT_LoCHO	FAT_LoCHO	HiCHO	PROT_LoCHO	FAT_LoCHO	HiCHO	PROT_LoCHO	FAT_LoCHO	0 h	2 h	2 h−0 h
Apparent total circulating energy, kcal/L ^1^	28.21 ^b^	27.77 ^b^	31.32 ^a^	29.52 ^b^	28.45 ^c^	34.16 ^a^	1.31 ^b^*	1.45 ^b^*	3.71 ^a^*	<0.001	<0.001	0.024
Glucose	13.08 ^a^	13.15 ^a^	11.78 ^b^	12.73 ^a^	12.36 ^a^	9.99 ^b^	−0.35 ^a^	−0.43 ^a,b^*	−1.46 ^b^*	<0.001	<0.001	0.084
β-hydroxybutyrate	0.13 ^b^	0.17 ^a^	0.19 ^a^	0.10 ^c^	0.22 ^b^	0.34 ^a^	−0.03 ^c^*	0.05 ^b^*	0.16 ^a^*	<0.001	<0.001	<0.001
Triglycerides (only glycerol)	0.95 ^a^	0.54 ^b^	0.54 ^b^	2.13 ^b^	2.36 ^b^	3.35 ^a^	1.18 ^c^*	1.84 ^b^*	2.82 ^a^*	<0.001	<0.001	<0.001
Triglycerides (only fatty acids)	19.31 ^a^	10.88 ^b^	10.90 ^b^	43.15 ^b^	47.79 ^b^	67.79 ^a^	23.84 ^c^*	37.22 ^b^*	57.19 ^a^*	<0.001	<0.001	<0.001
NEFAs	6.97 ^a,b^	7.64 ^a^	6.14 ^b^	3.82 ^b^	3.25 ^c^	4.81 ^a^	−3.15 ^b^*	−4.17 ^c^*	−1.16 ^a^*	0.011	<0.001	<0.001
Adjusted total fatty acids ^2^	59.56 ^c^	67.63 ^b^	70.46 ^a^	38.08 ^a^	34.02 ^a^	13.72 ^b^	−21.48 ^a^*	−31.73 ^b^*	−54.78 ^c^*	<0.001	<0.001	<0.001

Units are percentages of total apparent circulating energy unless otherwise indicated. Superscript letters differing within a row across foods for a timepoint denote mean values that are significantly different at *p* < 0.05 by Wilcoxon signed-rank test. Asterisks indicate significant differences between the 2 h and 0 h timepoints within an individual test food by Wilcoxon signed-rank test. Full data, including standard errors, are available in Appendix A. ^1^ Apparent circulating energy (kcal/L) = glucose energy (kcal/L) + β–hydroxybutyrate energy (kcal/L) + triglycerides energy (kcal/L) + adjusted fatty acids energy (kcal/L); see Methods. ^2^ Adjusted fatty acids (kcal/L) = total fatty acids (kcal/L) − triglycerides energy [fatty acids only] (kcal/L) − NEFA energy (kcal/L); see Methods. NEFAs—non-esterified fatty acids.

**Table 5 metabolites-14-00373-t005:** Effects of test foods on postabsorptive and postprandial levels of catabolic, anabolic, and signaling-type metabolites.

	0 h			2 h			2 h−0 h			Mixed Model *p* Value (Overall Food Effect)		
Analyte	HiCHO	PROT_LoCHO	FAT_LoCHO	HiCHO	PROT_LoCHO	FAT_LoCHO	HiCHO	PROT_LoCHO	FAT_LoCHO	0 h	2 h	2 h−0 h
**Catabolic**												
Total amino acids	−4.76 ^a,b^	−5.87 ^b^	−4.02 ^a^	1.80 ^c^	14.29 ^a^	8.06 ^b^	6.55 ^c^*	20.16 ^a^*	12.09 ^b^*	0.003	<0.001	<0.001
Total indoles	4.09 ^a^	−10.14 ^b^	−7.85 ^b^	5.09 ^a^	−5.77 ^b^	−7.05 ^b^	1.00 ^b^	4.38 ^a^*	0.80 ^b^	<0.001	<0.001	0.022
Total phenols	5.28 ^a^	−51.46 ^c^	−20.67 ^b^	11.33 ^a^	−35.27 ^c^	−5.58 ^b^	6.05 ^b^	16.19 ^a^*	15.09 ^a^*	<0.001	<0.001	0.015
Total MDAGs	−24.52 ^a^	−43.92 ^b^	−34.42 ^a^	11.69 ^a,b^	6.58 ^b^	15.15 ^a^	36.21 ^b^*	50.50 ^a^*	50.49 ^a,b^*	<0.001	0.018	0.022
Total acylcarnitines	−28.86 ^c^	−8.71 ^b^	8.40 ^a^	−34.50 ^c^	−25.14 ^b^	30.94 ^a^	−5.64 ^b^*	−16.43 ^c^*	22.54 ^a^*	<0.001	<0.001	<0.001
Total β-hydroxy FAs	−1.41 ^c^	0.87 ^b^	2.82 ^a^	−5.69 ^b^	−6.45 ^b^	6.71 ^a^	−4.28 ^b^*	−7.32 ^c^*	3.89 ^a^*	<0.001	<0.001	<0.001
Total α-hydroxy FAs	−3.87	−4.31	−3.38	2.00 ^a^	−1.41 ^b^	1.46 ^a^	5.25 *	2.18 *	4.10 *	0.679	0.009	0.181
Total ω-carboxy FAs	0.71 ^a^	−0.81 ^a,b^	−1.00 ^b^	0.24 ^b^	−3.36 ^c^	7.93 ^a^	−0.46 ^b^	−2.55 ^b^*	8.94 ^a^*	0.099	<0.001	<0.001
**Anabolic**												
Total GPCs	−2.25 ^b^	0.11 ^a^	1.43 ^a^	2.63 ^a^	0.52 ^b^	1.71 ^a,b^	4.88 ^a^*	0.41 ^b^	0.29 ^b^	0.001	0.062	<0.001
Total GPEs	−2.16 ^a^	−6.75 ^b^	−10.77 ^c^	6.10 ^a^	1.90 ^c^	3.40 ^b^	8.26 ^b^*	8.65 ^b^*	14.17 ^a^*	<0.001	<0.001	<0.001
Total SPHINGs	−11.18 ^b^	−8.69 ^b^	7.26 ^a^	−6.25 ^b^	−2.82 ^b^	7.43 ^a^	4.93 ^a^*	5.87 ^a^*	0.16 ^b^	<0.001	<0.001	<0.001
**Signaling**												
Total acylcholines	−3.07 ^c^	−1.34 ^b^	3.94 ^a^	5.03 ^a^	−8.20 ^c^	−0.95 ^b^	8.10 ^a^*	−6.86 ^b^*	−4.89 ^b^*	<0.001	<0.001	<0.001
Total acylethanolamides	0.04 ^b^	−0.07 ^b^	0.73 ^a^	0.17 ^a^	−0.83 ^b^	−0.54 ^b^	0.13 ^a^	−0.77 ^a,b^*	−1.27 ^b^*	0.049	0.011	0.024
Total acyl amino acids	−6.87 ^b^	−7.68 ^b^	−4.36 ^a^	−6.90 ^b^	−10.81 ^c^	5.43 ^a^	−0.03 ^b^	−3.13 ^b^	9.79 ^a^*	0.049	<0.001	<0.001
**Glycation**												
Total AGEs	−6.36 ^a^	−12.99 ^c^	−10.60 ^b^	3.72 ^b^	7.70 ^a^	6.92 ^a^	10.09 ^c^*	20.69 ^a^*	17.52 ^b^*	<0.001	<0.001	<0.001

Values are log_2_ median-centered relative fold values. Superscript letters differing within a row across foods for a timepoint denote mean values that are significantly different at *p* < 0.05 by Wilcoxon signed-rank test. Asterisks indicate significant differences between the 2 h and 0 h timepoints within an individual test food by Wilcoxon signed-rank test. Full data, including standard errors, are available in Appendix A, and full data for all constituent members of these classes are in Appendix A. AGEs—advanced glycation end products; FAs—fatty acids; GPCs, glycerophosphocholines; GPEs, glycerophosphoethanolamines; MDAGs—mono- and diacylglycerides; SPHINGs—sphingolipids.

**Table 6 metabolites-14-00373-t006:** Effects of test foods on postabsorptive and postprandial levels of serum total fatty acids and ratios for elongation and desaturation fatty acids.

	0 h			2 h			2 h−0 h			Mixed Model *p* Value (Overall Food Effect)		
Analyte	HiCHO	PROT_LoCHO	FAT_LoCHO	HiCHO	PROT_LoCHO	FAT_LoCHO	HiCHO	PROT_LoCHO	FAT_LoCHO	0 h	2 h	2 h−0 h
**Fatty acid (μM)**												
8:0	0.00	0.00	0.24	0.00	0.00	128.61	0.00	0.00	128.37	NA	NA	NA
10:0	0.76	0.17	12.38	0.00	0.00	184.68	−0.76	−0.17	172.30	NA	NA	NA
12:0	4.68 ^a^	3.74 ^a,b^	3.36 ^b^	3.45 ^a^	0.29 ^b^	0.30 ^b^	−1.24 ^a^	−3.45 ^b^*	−3.05 ^b^*	0.270	<0.001	0.031
14:0	22.97 ^a^	15.25 ^b^	15.22 ^b^	16.45 ^a^	11.18 ^b^	16.59 ^a^	−6.52 ^c^*	−4.07 ^b^*	1.36 ^a^	<0.001	<0.001	<0.001
16:0	1440.90 ^b^	1495.31 ^a^	1497.13 ^a^	1468.95 ^b^	1520.43 ^b^	1637.77 ^a^	28.05 ^b^	25.11 ^b^	140.64 ^a^*	0.087	<0.001	<0.001
16:1	87.90 ^b^	96.29 ^a^	89.68 ^b^	72.70 ^c^	80.02 ^b^	107.12 ^a^	−15.20 ^b^*	−16.27 ^b^*	17.45 ^a^*	0.135	<0.001	<0.001
18:0	2624.17 ^b^	2563.46 ^b^	3300.45 ^a^	2784.64 ^b^	2679.67 ^b^	3449.60 ^a^	160.47 *	116.21 *	149.14 *	<0.001	<0.001	0.461
18:1n9	836.73	831.30	869.30	770.47 ^b^	752.20 ^b^	963.54 ^a^	−66.26 ^b^*	−79.09 ^b^*	94.24 ^a^*	0.190	<0.001	<0.001
18:2n6	1960.81 ^a^	1571.01 ^c^	1773.10 ^b^	2062.00 ^a^	1632.24 ^c^	1952.81 ^b^	101.20 ^b^*	61.23 ^b^*	179.70 ^a^*	<0.001	<0.001	<0.001
18:3n3	53.75 ^c^	63.61 ^b^	91.65 ^a^	49.02 ^c^	65.42 ^b^	133.12 ^a^	−4.73 ^c^*	1.81 ^b^	41.48 ^a^*	<0.001	<0.001	<0.001
18:3n6	11.07 ^a^	10.13 ^b^	9.83 ^b^	9.60 ^b^	10.40 ^a^	11.09 ^a^	−1.47 ^b^*	0.28 ^a^	1.26 ^a^*	0.004	0.003	<0.001
20:2n6	29.59 ^a^	21.03 ^b^	21.02 ^b^	30.69 ^a^	19.34 ^b^	19.78 ^b^	1.10 ^a^	−1.68 ^b^*	−1.24 ^b^	<0.001	<0.001	0.021
20:3n6	106.35 ^b^	104.58 ^b^	164.78 ^a^	117.71 ^b^	106.10 ^c^	166.79 ^a^	11.36 ^a^*	1.52 ^b^	2.02 ^b^	<0.001	<0.001	<0.001
20:4n6	1825.26 ^b^	2080.74 ^a^	2050.25 ^a^	1941.51 ^b^	2131.57 ^a^	2083.86 ^a^	116.25 ^a^*	50.83 ^b^*	33.61 ^b^*	<0.001	<0.001	0.028
20:5n3	28.31 ^c^	35.15 ^b^	193.43 ^a^	26.60 ^c^	34.38 ^b^	196.06 ^a^	−1.71 ^b^*	−0.77 ^a,b^	2.63 ^a^	<0.001	<0.001	0.022
22:4n6	90.44 ^b^	117.25 ^a^	46.15 ^c^	93.51 ^b^	115.97 ^a^	46.03 ^c^	3.06 ^a^*	−1.28 ^b^	−0.12 ^b^	<0.001	<0.001	0.050
22:5n3	233.75 ^b^	187.68 ^c^	256.40 ^a^	243.24 ^a^	185.01 ^b^	252.18 ^a^	9.48 ^a^*	−2.67 ^b^	−4.22 ^b^	<0.001	<0.001	<0.001
22:6n3	101.38 ^b^	85.59 ^c^	273.03 ^a^	100.44 ^b^	85.26 ^c^	273.08 ^a^	−0.94	−0.33	0.06	<0.001	<0.001	0.949
**Enzyme Ratios**												
SCD1 (d9) (16:1/16:0)	0.061 ^a,b^	0.064 ^a^	0.060 ^b^	0.050 ^c^	0.053 ^b^	0.065 ^a^	−0.011 ^b^*	−0.012 ^b^*	0.005 ^a^*	0.221	<0.001	<0.001
SCD1 (d9) (18:1/18:0)	0.32 ^a^	0.33 ^a^	0.27 ^b^	0.28	0.28	0.28	−0.04 ^b^*	−0.04 ^b^*	0.01 ^a^*	<0.001	0.756	<0.001
d6 desaturase (18:3n6/18:2n6)	0.0057 ^b^	0.0064 ^a^	0.0057 ^b^	0.0047 ^c^	0.0064 ^a^	0.0057 ^b^	−0.001 ^b^*	0.00 ^a^	0.00 ^a^	<0.001	<0.001	0.001
d5 desaturase (20:4n6/20:3n6)	18.13 ^b^	20.42 ^a^	12.96 ^c^	17.48 ^b^	20.57 ^a^	12.95 ^c^	−0.65 ^b^*	0.15 ^a^	−0.01 ^a^	<0.001	<0.001	<0.001
Elongase Elovl-6 (18:0/16:0)	1.82 ^b^	1.71 ^c^	2.21 ^a^	1.90 ^b^	1.76 ^c^	2.11 ^a^	0.07 ^a^*	0.05 ^a^*	−0.09 ^b^*	<0.001	<0.001	<0.001
Elongase Elovl-5 (20:3n6/18:3n6)	9.79 ^c^	10.57 ^b^	17.82 ^a^	12.53 ^b^	10.47 ^c^	15.57 ^a^	2.74 ^a^*	−0.11 ^b^	−2.25 ^b^	<0.001	<0.001	<0.001
Overall Elongation ((18:0 + 18:1)/16:0)	2.40 ^b^	2.27 ^c^	2.79 ^a^	2.42 ^b^	2.26 ^c^	2.70 ^a^	0.02 ^a^	−0.01 ^b^*	−0.09 ^c^*	<0.001	<0.001	<0.001

Superscript letters differing within a row across foods for a timepoint denote mean values that are significantly different at *p* < 0.05 by Wilcoxon signed-rank test. Asterisks indicate significant differences between the 2 h and 0 h timepoints within an individual test food by Wilcoxon signed-rank test. Full data, including standard errors, are available in Appendix A. Elovl—elongation of very-long-chain fatty acids; SCD1—stearoyl CoA desaturase.

**Table 7 metabolites-14-00373-t007:** Effects of test foods on postabsorptive and postprandial levels of hunger and satiety hormones.

	0 h			2 h			2 h−0 h			Mixed Model *p* Value (Overall Food Effect)		
Hormone	HiCHO	PROT_LoCHO	FAT_LoCHO	HiCHO	PROT_LoCHO	FAT_LoCHO	HiCHO	PROT_LoCHO	FAT_LoCHO	0 h	2 h	2 h−0 h
Insulin	272.00	292.52	346.91	549.73 ^b^	709.57 ^a^	602.65 ^b^	277.73 ^a,b^*	417.05 ^a^*	255.74 ^b^*	0.306	0.006	0.112
Glucagon	87.23	83.74	90.90	123.68 ^c^	239.52 ^a^	200.93 ^b^	36.45 ^c^*	155.79 ^a^*	110.03 ^b^*	0.405	<0.001	<0.001
Insulin/Glucagon (pM/pM)	1.89	2.11	2.13	2.78 ^a^	1.95 ^a,b^	1.82 ^b^	0.89 ^a^*	−0.20 ^b^	−0.30 ^b^	0.442	0.014	0.002
GIP	8.15 ^a^	6.48 ^b^	8.45 ^a,b^	168.19 ^b^	144.93 ^c^	217.65 ^a^	160.03 ^b^*	138.44 ^b^*	209.20 ^a^*	0.046	0.001	0.079
GLP-1	3.66	2.41	9.03	26.00 ^b^	33.90 ^b^	51.28 ^a^	22.34 ^b^*	31.48 ^a^*	42.25 ^a^*	0.081	<0.001	0.115
PP	214.87	125.65	178.40	802.89 ^a^	811.71 ^a^	649.14 ^b^	588.02 ^a,b^*	686.06 ^a^*	470.73 ^b^*	0.091	<0.001	0.018
PYY	154.54 ^a^	107.08 ^b^	162.22 ^a^	264.77 ^a,b^	247.22 ^b^	323.78 ^a^	110.23 *	140.14 *	161.56 *	0.020	0.018	0.449
Leptin	3241.29 ^a^	2113.62 ^b^	3513.98 ^a^	2976.98	2662.05	3001.70	−264.31 ^b^	548.43 ^a^*	−512.28 ^b^	<0.001	0.464	<0.001
Ghrelin	728.10 ^b^	797.27 ^b^	1150.16 ^a^	628.35 ^a^	326.21 ^b^	585.90 ^a^	−99.74 ^a^	−471.06 ^b^*	−564.26 ^b^*	<0.001	<0.001	<0.001
Leptin/Ghrelin (pM/pM)	14.74 ^a^	8.40 ^b^	7.55 ^b^	18.73 ^b^	31.12 ^a^	15.68 ^b^	3.99 ^c^	15.82 ^a^*	8.13 ^b^*	0.005	0.061	0.009

Superscript letters differing within a row across foods for a timepoint denote mean values that are significantly different at *p* < 0.05 by Wilcoxon signed-rank test. Asterisks indicate significant differences between the 2 h and 0 h timepoints within an individual test food by Wilcoxon signed-rank test. Units of measurement are pg/mL except where otherwise indicated. The following masses were used: insulin—5.8 kDa; glucagon—3.485 kDa; leptin—16 kDa; ghrelin—36 kDa (average of 32 and 41 kDa variants). Full data, including standard errors, are available in Appendix A. GIP—gastric inhibitory peptide; GLP-1—glucagon-like peptide-1; PP—pancreatic polypeptide; PYY—peptide tyrosine tyrosine.

## Data Availability

The original contributions presented in the study are included in the article/Appendix A; further inquiries can be directed to the corresponding author.

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
