# Peer review of "Replacement of Dietary Carbohydrate with Protein versus Fat Differentially Alters Postprandial Circulating Hormones and Macronutrient Metabolism in Dogs"

_metabolites, 2024, doi:10.3390/metabo14070373_

Round 1

Reviewer 1 Report

Comments and Suggestions for Authors

The study is very interesting and presents original and valuable data.

As a whole the introduction provides the necessary background and what is new in the area, however, the aim of the study is not very clearly stated in this section.

The material and method section is rather concise and the author refers constantly to a previous research. In my opinion, a brief description of each individual analysis will improve the section.

The results are clearly described and the number of tables is adequate.  The discussion is professionally done. The conclusions are derived from the results.

Author Response

Comment 1: The study is very interesting and presents original and valuable data.

Response 1: Thank you for your perspective and generous comment. I am grateful to the Reviewer for their time and insight in reviewing this manuscript draft. The comments are much appreciated, and I have sought to address both of the points raised so as to improve the quality of the manuscript by making more apparent the overarching aim of the investigation as well as the methods employed.

Comment 2: As a whole the introduction provides the necessary background and what is new in the area, however, the aim of the study is not very clearly stated in this section.

Response 2: Thank you, this point is well taken, and it was not clear what the specific aim of the investigation was in the draft. New text has been added to the draft toward the end of the Introduction that outline the general and specific aim of the trial and how this might contribute to enhanced understanding of canine physiology (lines 91-96, 104-105).

Comment 3: The material and method section is rather concise and the author refers constantly to a previous research. In my opinion, a brief description of each individual analysis will improve the section.

Response 3: Thank you for this comment. I now better appreciate how this repeated reference to the preceding publication places an undue burden on the Reviewer and readers to seek out and parse through a separate publication. The Methods section has now been updated with new text that briefly describes the methodology employed (lines 128-133, 189-197, 203-222), so hopefully this will be at an acceptable level of detail to allow readers to understand the methods without being redundant with the preceding publication where those methods are described in more detail.

Comment 4: The results are clearly described and the number of tables is adequate.  The discussion is professionally done. The conclusions are derived from the results.

Response 4: Thank you for your comment and I hope this section may be of value to readers in integrating existing understandings with these new results.

Reviewer 2 Report

Comments and Suggestions for Authors

The work submitted for review entitled "Replacement of Dietary Carbohydrate With Protein Versus Fat Differentially Alters Postprandial Circulating Hormones and Macronutrient Metabolism in Dogs" is a very interesting, well-designed and conducted study.

The Author used self-citation of four works, to which the reader is referred to in order learn the methodology of markings described in the assessed work.

My comments are:

What does the Author mean when he writes about the "circulating energy"?

Could the Author explain the reason for the increased albumin level in the FAT_LoCHO diet?

For a clearer description, can the Author explain what exactly he means by "clinical analyses"?

Author Response

Comment 1: The work submitted for review entitled "Replacement of Dietary Carbohydrate With Protein Versus Fat Differentially Alters Postprandial Circulating Hormones and Macronutrient Metabolism in Dogs" is a very interesting, well-designed and conducted study.

Response 1: Thank you for your consideration; very much appreciated. I am grateful to the Reviewer for taking time to consider this manuscript draft and provide their review. The comments are certain to improve the readability and rigor of the manuscript. Herein I have tried to address the points raised.

Comment 2: The Author used self-citation of four works, to which the reader is referred to in order learn the methodology of markings described in the assessed work.

Response 2: Thank you for this comment, I now better appreciate how this continual reference to the preceding publication places an undue burden on the Reviewer and readers to seek out and parse through a separate publication. The Methods section has now been updated with new text that briefly describes the methodology employed (lines 128-133, 189-197, 203-222). Hopefully this will be at an acceptable level of detail to allow readers to understand the methods without being redundant with the preceding publication where those methods are described in more detail.

Regarding the self-citations: Please note that only one of those citations was a citation referencing methodology. And, as indicated above, there is new text providing more detail in the Methods section so that the Reviewer and readers will not need to acquire and parse through a separate reference to understand how the analyses were performed.

Regarding the other three self-citations: Please note that the remaining self-citations are only but three references (#96-98) among 15 total references (references #95-110) that are in the Discussion section related to amino acid putrefaction by gut microbial processes. These references are instrumental in developing an integrated perspective on how the current results provide new observations that higher protein foods do not necessarily lead to higher levels of circulating putrefactive postbiotic products of colonic microbes.

Comment 3: My comments are:

What does the Author mean when he writes about the "circulating energy"?

Response 3: Thank you for pointing out the obscurity of “circulating energy” as was defined in the original submission. I am grateful to get the opportunity to better define this metric as I believe it to be an important finding from this investigation. New text has been added to the Methods section to fully define the term circulating energy and how it was derived from biochemical measurements so that the reader does not need to access any external literature to understand the term (lines 196-197, 203-222). The abstract has been updated to define the term circulating energy at a high level (lines 13-16) so a reader of the abstract alone will grasp what the term means.

Comment 4: Could the Author explain the reason for the increased albumin level in the FAT_LoCHO diet?

Response 4: Great question! Thank you for inquiring. I thought a lot about that during development of the manuscript and have previously explored some explanations but do not have a definitive answer. Prompted by your question, and imagining that others will be also interested in this aspect of the current manuscript, the manuscript has been updated with additional text and data in the Results (lines 310-330) and Discussion sections (lines 955-973) describing the analyses and hypotheses that came out of the considerations of what is driving albumin level differences across the foods.

Comment 5: For a clearer description, can the Author explain what exactly he means by "clinical analyses"?

Response 5: Thanks for this comment; I now realize it is an awkward term. Rather than “clinical”, a more appropriate term is “biochemical” since the measures reported here in the relevant sections are standard biochemical analytes routinely assessed as markers of metabolic physiology. The manuscript has been updated to change all instances of “clinical” to “biochemical.”